# Provable Training Data Identification for Large Language Models

Zhenlong Liu [1 2]   Hao Zeng [1]   Weiran Huang [2 3]   Hongxin Wei [1]

## Abstract

Identifying training data of large-scale models is critical for copyright litigation, privacy auditing, and ensuring fair evaluation. However, existing works typically treat this task as an instance-wise identification without controlling the error rate of the identified set, which cannot provide statistically reliable evidence. In this work, we formalize training data identification as a set-level inference problem and propose Provable Training Data Identification (**PTDI**), a distribution-free approach that enables provable and strict false identification rate control. Specifically, our method computes conformal p-values for each data point using a set of known unseen data and then develops a novel Jackknife-corrected Beta boundary (JKBB) estimator to estimate the training-data proportion of the test set, which allows us to scale these p-values. By applying the Benjamini–Hochberg (BH) procedure to the scaled p-values, we select a subset of data points with provable and strict false identification control. Extensive experiments across various models and datasets demonstrate that PTDI achieves higher power than prior methods while strictly controlling the FIR. Our implementation code is available at https://github.com/zhenlong-liu/Provable_Training_Data_Identification

## 1. Introduction

The extensive deployment of machine learning models has driven an ongoing demand for large-scale datasets, which has raised significant legal challenges, including copyright disputes (Bartz et al., 2024; Disney Enterprises, Inc., 2025), data privacy concerns (European Parliament & Council of the European Union, 2016; California State Assembly, 2018), and issues of data contamination from evaluation benchmarks (Sainz et al., 2023; Balloccu et al., 2024). These concerns raise the importance of identifying a specific, well-defined set of data allegedly used in training. For instance, Strike 3 alleges that Meta infringed on at least 2,396 of its copyrighted films in its lawsuit (Strike 3 Holdings, LLC and Counterlife Media, LLC, 2025), a claim with potential statutory damages exceeding $350 million. To resolve such high-stakes disputes, claims must be supported by credible evidence that strictly controls the risk of false positives. This underscores the need for methods that provide rigorous statistical guarantees for identifying training data.

To this end, prior studies (Shi et al., 2024; Li* et al., 2024; Zhang et al., 2025a) have developed various methods to detect training data from large language models (LLMs) and vision-language models (VLMs). These methods typically compute a detection score (e.g., perplexity or entropy) for a given sample and infer whether it was used in training via level-set estimation, without providing any theoretical guarantee. Moreover, due to the scale and complexity of LLMs/VLMs, instance-wise inference is statistically intractable for providing rigorous evidence for the identified training data (Zhang et al., 2025b). In high-stakes settings such as legal disputes or privacy auditing, this lack of error control limits their practice as credible evidence. This motivates us to design a provable method for training data identification that provides rigorous control over the false identification rate (FIR) of the selected set.

In this work, we formally define the problem of training data identification by shifting the focus from instance-wise classification to set-level identification, thereby enabling provable error control. For this problem, we introduce Provable Training Data Identification (**PTDI**), a distribution-free method for training-data identification with provable FIR control. Our approach first computes detection scores for each data point and then constructs valid p-values via conformal inference, assuming the calibration set and test non-members are from the same distribution. To improve power, we further estimate the training-data proportion in the test set via a Jackknife-corrected Beta boundary (JKBB) estimator, and use it to rescale the p-values before applying the Benjamini–Hochberg procedure (Benjamini & Hochberg, 1995; Benjamini & Yekutieli, 2001). We provide rigorous

---

[1]Department of Statistics and Data Science, Southern University of Science and Technology [2]Shanghai Innovation Institute [3]School of Computer Science, Shanghai Jiao Tong University. Correspondence to: Hongxin Wei <weihx@sustech.edu.cn>.

*Proceedings of the 43$^{rd}$ International Conference on Machine Learning*, Seoul, South Korea. PMLR 306, 2026. Copyright 2026 by the author(s).

theoretical guarantees that this data-dependent scaling preserves valid FIR control, while substantially increasing the power. The resulting method is compatible with both black-box outputs and white-box gradient-based scores, and is complementary to existing detection methods, making it a practical tool for training data identification.

We empirically validate our method through extensive experiments across diverse settings, including pre-training and fine-tuning paradigms for LLMs and VLMs, evaluated on multiple benchmark datasets (e.g., WikiMIA (Shi et al., 2024), ArXivTection (Duarte et al., 2024), VL-MIA/Flickr and VL-MIA/DALL-E (Li et al., 2024c)). Across all settings, our method reliably identifies training data while maintaining error control, providing strong empirical support for our theoretical guarantees. For instance, on WikiMIA with Pythia-1.4B (Biderman et al., 2023) at a target FIR of 0.05, our method achieves an empirical FIR of 0.0494, whereas the approach of Hu et al. (2025) yields 0.1311. We further demonstrate that our p-value scaling procedure improves power over the vanilla conformal approach. Specifically, on WikiMIA with GPT-NeoX-20B (Black et al., 2022) at a target FIR of 0.5, our method improves power from 0.5177 to 0.8618 using the MIN-K% detection score (Shi et al., 2024). In summary, PTDI provides a practical and statistically reliable tool for training data identification in high-stakes scenarios.

We summarize our contributions as follows:

1. We formulate training-data identification as a set-level inference problem and propose **PTDI**, a distribution-free method that achieves rigorous FIR control. By leveraging conformal p-values with a novel JKBB estimator based on non-member calibration data, our method applies to arbitrary detection scores. Extensive experiments demonstrate our method achieves consistent and reliable error control.

2. We establish theoretical guarantees for PTDI by showing consistency of the JKBB estimator at the boundary, which supports the validity of the scaled p-value procedure and ensures FIR control.

3. We extend our method to scenarios where a small set of member data is available. We propose an adjusted moment estimator that leverages this information to provide a consistent estimate of data usage proportion, further enhancing identification power while maintaining the theoretical guarantee.

## 2. Background

**Training data detection.** Given a data point $X$ and a target model $\theta$ trained on dataset $\mathcal{D}_{\text{train}}$, training data detection aims to detect whether $X$ is a part of the training set $\mathcal{D}_{\text{train}}$.

This problem is an instance of the membership inference attacks (MIAs) (Shokri et al., 2017), but generally applied to the LLM/VLM scenarios (Carlini et al., 2021; Zhang et al., 2024; Shi et al., 2024; Li et al., 2024c). This task is typically formulated as a binary classification problem, where the predicted label $\widehat{M} \in \{0, 1\}$ indicates whether $X$ is predicted as a training sample ($\widehat{M} = 1$) or not ($\widehat{M} = 0$). Formally, this prediction is made through level-set estimation:

$$\widehat{M} = \mathbb{1}\{T(X; \theta) \leqslant \tau\}, \tag{1}$$

where $T(X; \theta)$ denotes the detection score (e.g., perplexity) calculated from the model $\theta$, and $\tau$ is a threshold determined by a validation set. By convention, a lower detection score $T$ suggests $X$ is more likely to be trained by the target model.

To provide a concrete understanding of the detection score $T(X; \theta)$, we now introduce a widely-used example. For LLMs, where a data point $X = \{x_1, \ldots, x_L\}$ is a text sequence, a common detection score is perplexity (Li, 2023):

$$\text{Perplexity}(X; \theta) = \exp[-\sum_{i=1}^{L} \log p_\theta(x_i \mid x_{<i})], \tag{2}$$

where $x_{<i} = (x_1, \ldots, x_{i-1})$ and $p_\theta(x_i \mid x_{<i})$ denotes the conditional probability of token $x_i$ given its preceding tokens. A lower perplexity suggests the sequence is more familiar to the model, which is consistent with the sequence being drawn from the model's training distribution. In the Section 4.1, we will further consider a variety of detection scores, including Zlib (Carlini et al., 2021), M-Entropy (Song & Mittal, 2021), MIN-K% (Shi et al., 2024) for LLMs, and MaxRényi-K% (Li et al., 2024c) for VLMs.

The above-mentioned training data detection method can only provide a metric for instance-wise classification, failing to provide rigorous guarantees needed in real-world scenarios. Furthermore, Zhang et al. (2025a) argue that proving whether a model was trained on a specific data point is intractable due to limited access to the training process and prohibitive training costs. This motivates us to develop a method for identifying training data with provable evidence.

## 3. Our proposed method

### 3.1. Problem formulation

To address the limitation, we shift our focus from instance-wise decisions to set-level inference. In many practical applications, the objective is not merely to classify single data points but to identify a **subset of members** from a larger collection. For instance, in data contamination research, identifying and removing sets of contaminated data points is crucial for fair model evaluation (Dong et al., 2024; Zhu et al., 2024b; Zhao et al., 2024; Gao et al., 2024). Similarly, in copyright litigation, claimants must provide a specific list

of infringed works, where the ability to produce a credible set of evidence can have significant financial implications Therefore, providing a credible set containing training data is crucial for the training data discovery. This naturally leads to a multiple-hypothesis testing formulation, where each candidate sample corresponds to a hypothesis about its membership status, and guarantees must be provided at the level of the selected set rather than individual predictions.

In this work, we formally define the problem of provable **training data identification**, where the objective is to construct a selection set from the test data that contains a provable proportion of true member samples. Suppose we have access to a target model $\theta$, a calibration set $\mathcal{D}_{\text{cal}}$ of size $n$ and a test set $\mathcal{D}_{\text{test}} = \{X_{n+j}\}_{j=1}^{m}$ consisting of candidate training samples. Here, $M_i \in \{0, 1\}$ denotes the true membership label, where $M_i = 1$ indicates that $X_i$ was used to train $\theta$. Then we formulate the problem within the framework of multiple hypothesis testing:

$$H_{0,j} : M_{n+j} = 0, \quad j = 1, \ldots, m, \quad (3)$$

Our goal is to select a subset of indices $\mathcal{S} \subseteq \{1, \ldots, m\}$ from $\mathcal{D}_{\text{test}}$ such that the false identification rate (FIR) is controlled at a user-specified level $\alpha \in (0, 1)$:

$$\text{FIR} = \mathbb{E}\left[ \frac{\sum_{j=1}^{m} \mathbb{1}\{M_{n+j} = 0, j \in \mathcal{S}\}}{\max(|\mathcal{S}|, 1)} \right] \leqslant \alpha, \quad (4)$$

where the quantity inside the expectation is the false identification proportion (FIP), representing the fraction of false identifications in the selected set $\mathcal{S}$. At this guarantee, we also desire $\mathcal{S}$ containing true training data points as much as possible, which is quantified by power:

$$\text{Power} = \mathbb{E}\left[ \frac{\sum_{j=1}^{m} \mathbb{1}\{j \in \mathcal{S}, M_{n+j} = 1\}}{\max(1, \sum_{j=1}^{m} \mathbb{1}\{M_{n+j} = 1\})} \right]. \quad (5)$$

It is worth noting that this formulation of training data identification is different from traditional membership inference. The latter focuses on instance-wise classification, and providing theoretical guarantees for this task against large models is often intractable (Zhang et al., 2025b). In contrast, our work shifts the focus to a set-level identification, thereby enabling rigorous statistical error control in practice. A detailed discussion is provided in Appendix G.

In this paper, we mainly discuss the scenario that the auditor is only able to source data that are confirmed **non-members** of the training set. The calibration set $\mathcal{D}_{\text{cal}} = \{(X_i, M_i)\}_{i=1}^{n}$ is constructed such that $M_i = 0$ for all $i$. This is a widely used assumption (Ye et al., 2022; Shi et al., 2024; Zhang et al., 2025a) since it can be satisfied by using data generated after the model's training cutoff date (e.g., recent news articles or photos) or by leveraging private, proprietary data not publicly accessible for web scraping

---

**Algorithm 1** Provable Training Data Identification(PTDI)

**Require:** Target model $\theta$, calibration data $\mathcal{D}_{\text{cal}}$, test data $\mathcal{D}_{\text{test}}$, FIR target $\alpha \in (0, 1)$, detection score function $T(\cdot)$, data usage proportion estimator $\mathcal{E}$.
1: Compute detection scores $T_i \leftarrow T(X_i; \theta)$ for all $X_i \in \mathcal{D}_{\text{cal}} \cup \mathcal{D}_{\text{test}}$.
2: Construct p-values $p_j$ as Equation (6) for $j = 1, \ldots, m$.
3: Obtain the data usage proportion estimate $\hat{\pi}_{\text{test}} \leftarrow \mathcal{E}(\mathcal{D}_{\text{cal}}, \mathcal{D}_{\text{test}})$
4: Compute scaled p-values: $\tilde{p}_j \leftarrow (1 - \hat{\pi}_{\text{test}})p_j$ for $j = 1, \ldots, m$.
5: Sort the scaled p-values: $\tilde{p}_{(1)} \leqslant \tilde{p}_{(2)} \leqslant \cdots \leqslant \tilde{p}_{(m)}$.
6: Find $k^* \leftarrow \max\{k \mid \tilde{p}_{(k)} \leqslant \frac{k}{m}\alpha\}$.
7: **if** $k^*$ exists **then**
8:     **return** $\mathcal{S} = \{j \mid \tilde{p}_j \leqslant \frac{k^*}{m}\alpha\}$
9: **else**
10:     **return** $\mathcal{S} = \emptyset$.
11: **end if**

---

(e.g., internal corporate documents or unreleased creative works). We proceed by constructing valid p-values for these hypotheses using the calibration set.

### 3.2. Provable Training Data Identification

To achieve the training–data identification objective in Section 3.1, we derive p-values that are valid under each null hypothesis. For notational convenience, let $T_i = T(X_i; \theta)$ for $i \in \{1, \ldots, n + m\}$. We next construct the conformal p-values (Vovk et al., 2003; 2005) for each test point as:

$$p_j = \frac{1 + \sum_{i=1}^{n} \mathbb{1}\{T_i \leqslant T_{n+j}\}}{n + 1}, \text{ for } j = 1, \ldots, m. \quad (6)$$

Conceptually, a smaller score $T_{n+j}$ indicates that $X_{n+j}$ is more likely a training member, resulting in a smaller p-value. To collectively test hypotheses for all test instances with controlled FIR, we employ the Benjamini-Hochberg (BH) procedure (Benjamini & Hochberg, 1995). However, the standard BH procedure is conservative as its theoretical FIR bound scales with the proportion of true null hypotheses (Storey, 2002). To improve the power, we introduce scaled p-values, which adjust for the estimated proportion of training data in the target set. The scaled p-value is defined as:

$$\tilde{p}_j = (1 - \hat{\pi}_{\text{test}})p_j, \text{ for } j = 1, \ldots, m. \quad (7)$$

where $\hat{\pi}_{\text{test}}$ is an estimate of $\pi_{\text{test}}$, the proportion of training data in the test set (i.e., $\pi_{\text{test}} = \Pr(M_{n+j} = 1)$). This estimate is obtained via a data usage proportion estimator $\mathcal{E}$ such that $\hat{\pi}_{\text{test}} = \mathcal{E}(\mathcal{D}_{\text{cal}}, \mathcal{D}_{\text{test}})$. We defer the implementation details of this estimator to Section 3.3.

With these scaled p-values, we then run the BH procedure to obtain the set of identified training data. Specifically, let

$\tilde{p}_{(1)} \leqslant \tilde{p}_{(2)} \leqslant \cdots \leqslant \tilde{p}_{(m)}$ denote the sorted scaled p-values, the final set is:

$$\mathcal{S} = \{j \mid \tilde{p}_j \leqslant \frac{k^*}{m}\alpha\}, \text{ where } k^* = \max\{k \mid \tilde{p}_{(k)} \leqslant \frac{k}{m}\alpha\}. \tag{8}$$

This procedure determines a data-dependent significance threshold by identifying the largest p-value $\tilde{p}_{(k^*)}$, and then identifies all data points with scaled p-values below this adaptive threshold as significant. The full procedure is detailed in Algorithm 1. To complete this procedure, we next introduce a practical method for estimating $\pi_{\text{test}}$.

### 3.3. Estimate data usage proportion

In this subsection, we detail a boundary density estimation implementation of the data usage proportion estimator $\mathcal{E}$ required by our main procedure in Algorithm 1. The resulting estimate $\hat{\pi}_{\text{bdy}}$ will be used as $\hat{\pi}_{\text{test}}$ to scale the p-values.

Recall that the p-values $\{p_j\}_{j=1}^m$ in Equation (6) are computed by comparing test scores against calibration scores from $\mathcal{D}_{\text{cal}}$. This conformal framework leads to a distribution-free property: regardless of the underlying distribution of the raw scores $T$ (which may vary across different samples or domains), the p-values under the null hypothesis ($M = 0$) are guaranteed to be super-uniformly distributed on $[0, 1]$. This allows us to map heterogeneous detection scores into a unified p-value space for collective analysis. Let $f_{\text{test}}(p)$ denote the marginal density of these p-values across the entire test set. Since each test point is either a non-member ($M_{n+j} = 0$) or a member ($M_{n+j} = 1$), $f_{\text{test}}$ admits a two-component mixture decomposition:

$$f_{\text{test}}(p) = (1-\pi_{\text{test}})\, f(p \mid M = 0) + \pi_{\text{test}}\, f(p \mid M = 1), \tag{9}$$

where $\pi_{\text{test}} = \Pr(M_{n+j} = 1)$ is the unknown training data proportion in the test set.

A key observation is that under the null hypothesis, conformal p-values are approximately uniform on $[0, 1]$, hence $f(p \mid M = 0) = 1$. Moreover, for a broad class of alternatives, the density of member p-values is suppressed near the upper boundary $p = 1$, i.e., $f(p \mid M = 1)$ is small in a neighborhood of 1. Consequently, the boundary value of the mixture density identifies the non-member proportion:

$$f_{\text{test}}(1) \gtrsim 1 - \pi_{\text{test}}. \tag{10}$$

Therefore, estimating $\pi_{\text{test}}$ reduces to estimating the p-value density at the boundary $p = 1$.

**Jackknife-corrected Beta boundary estimator.** To estimate the data usage proportion, we rely on accurately estimating the boundary density $f_{\text{test}}(1)$. However, the widely used Storey's estimator (Storey, 2002) estimates this boundary density using an average over a fixed tail interval. Since

valid detection scores typically yield a p-value density with a downward trend near the boundary, such averaging inherently leads to an inflated estimate of the null proportion (Neuvial, 2013), rendering the subsequent detection procedure **overly conservative**. To address this and recover the statistical power lost to this estimation bias, we propose using a boundary-adaptive kernel approach that explicitly adjusts for local curvature.

Specifically, we adopt Beta kernel density estimation (Chen, 1999), whose kernels are naturally supported on $[0, 1]$. Let $b > 0$ denote a bandwidth parameter, the Beta kernel evaluated at the boundary $p = 1$ admits the closed form

$$K_{1,b}(t) = \left(\frac{1}{b} + 1\right) t^{1/b}, \qquad t \in [0, 1]. \tag{11}$$

Given the test p-values $\{p_j\}_{j=1}^m$, a naive boundary density estimator is obtained by averaging $K_{1,b}(p_j)$. However, boundary kernel estimators typically suffer from a first-order bias in the bandwidth $b$. For a sufficiently smooth density $f_{\text{test}}$, the expectation of the boundary Beta kernel estimator admits the expansion:

$$\mathbb{E}\left[\widehat{f}_b(1)\right] = f_{\text{test}}(1) + c_1 b + c_2 b^2 + O(b^3), \tag{12}$$

where $c_1$ and $c_2$ depend on local derivatives of $f_{\text{test}}$ at the boundary. To remove the leading $O(b)$ bias term, we employ a dual-kernel jackknife technique (Schucany & Sommers, 1977). Specifically, we form a linear combination of two boundary estimators with bandwidths $b$ and $\gamma b$ ($\gamma > 0$):

$$\widehat{f}_{\text{jk}}(1) = w_0\, \widehat{f}_b(1) + w_1\, \widehat{f}_{\gamma b}(1), \tag{13}$$

where the weights are chosen to satisfy the constraints $w_0 + w_1 = 1$ and $w_0 + \gamma w_1 = 0$, yielding

$$w_0 = \frac{\gamma}{\gamma - 1}, \qquad w_1 = -\frac{1}{\gamma - 1}. \tag{14}$$

We then define the final estimator $\hat{\pi}_{\text{bdy}}$ as the Jackknife-corrected Beta Boundary (JKBB) estimator:

$$\hat{\pi}_{\text{bdy}} = 1 - \widehat{f}_{\text{jk}}(1), \tag{15}$$

which is used as $\hat{\pi}_{\text{test}}$ in Equation (7) and Algorithm 1. The detailed implementation regarding the choice for the bandwidth parameter $b$ and hyperparameter $\gamma$ is provided in the Appendix C. Equipped with the boundary density estimator $\hat{\pi}_{\text{bdy}}$, we now establish the asymptotic FIR control of the resulting plug-in procedure:

**Theorem 3.1.** *Suppose the covariate of calibration set $\{X_i\}_{i=1}^n$ and the test set $\{X_{n+j}\}_{j=1}^m$ are i.i.d. Then for any $\alpha \in (0, 1)$, the selected set $\mathcal{S}$ obtained by Algorithm 1 satisfies FIR $\leqslant \alpha$. That is:*

$$\limsup_{m \to \infty} \text{FIR}_m \leqslant \alpha.$$

The corresponding proof is presented in Appendix B.2. For reference, we also provide the evaluation of the JKBB estimator in Appendix F.1. In the following section, we empirically evaluate the behavior of the proposed procedure and verify its FIR control in practice.

## 4. Experimental results

### 4.1. Setup

**Models** Our experiments cover a wide range of open-source models. For LLMs, we evaluate GPT-2 (Radford et al., 2019), GPT-Neo (Gao et al., 2020), GPT-NeoX-20B (Black et al., 2022), LLaMA-7B (Touvron et al., 2023), and Pythia (1.4B and 6.9B variants) (Biderman et al., 2023). For VLMs, we use LLaVA-1.5 (Liu et al., 2023) and MiniGPT-4 (Zhu et al., 2024a).

**Datasets.** We employ six common benchmark datasets for evaluation. For LLM pre-training, we use the WikiMIA (Shi et al., 2024) and ArxivTection (Duarte et al., 2024) datasets. For fine-tuning LLMs, we utilize XSum (Narayan et al., 2018) and BBC Real Time (Li et al., 2024a). In the vision-language domain, following previous work (Li et al., 2024c), we use the VL-MIA/Flickr and VL-MIA/DALL-E datasets. The details for our experiment are presented in Appendix E.

**Detection Scores.** We employ a diverse set of detection scores to evaluate the versatility of our method. For LLMs, we utilize Perplexity (Li, 2023), ratio of perplexity to zlib compression entropy (Zlib) (Carlini et al., 2021), the modified entropy (M-Entropy) (Song & Mittal, 2021), and MIN-K% (Shi et al., 2024), which focuses on tokens with minimum probabilities. For VLMs, we follow Li et al. (2024c) and use the MaxRényi-K% score.

### 4.2. Main results

**Our method provides reliable error control with various detection scores.** Our method is designed to be score-agnostic and can be readily combined with a wide range of existing training data detection methods. For LLMs, we evaluate our approach using several widely adopted scores, including Perplexity, Zlib, M-Entropy, and MIN-K%. As shown in Figure 1, our method consistently and strictly controls the FIR across all settings. This demonstrates that PTDI is complementary to existing detection scores, enabling them to support training data identification with rigorous error control. Additional results for VLMs are provided in Appendix F.2.

**Our method achieves valid error control compared to the knockoff inference-based training data detector (KTD).** We adopt the experimental setting of KTD (Hu et al., 2025), evaluating on GPT-2 (Radford et al., 2019), GPT-Neo (Gao

et al., 2020), and Pythia-1.4B (Biderman et al., 2023) across the WikiMIA, XSum, and BBC Real Time datasets. To ensure a fair comparison under a white-box assumption, we set our method's detection score $T(X)$ to be the knockoff statistic from KTD. This statistic is calculated as the difference between the $L_2$ norm of the model's gradient for an input $X$ and the average $L_2$ norm of the gradients for its synthetic knockoff samples. As demonstrated in Figure 3, our method consistently maintains the target FIR across all settings, whereas KTD fails to control the FIR on WikiMIA and XSum for certain values of $\alpha$. For a complete analysis, we compare the power on BBC Real Time under conditions where KTD successfully controls the FIR ($\alpha \geqslant 0.1$), with results shown in Figure 2. The comparison reveals that our method achieves superior power on GPT-2. In summary, our method not only guarantees strict FIR control but also demonstrates superior power.

### 4.3. Ablation Study

**Effect of selection procedures on power.** To analyze the effect of different selection procedures, we compare our method with several alternative approaches, including BH-Storey (Storey, 2002), BKY (Benjamini et al., 2006), Quantile-BH (Benjamini & Hochberg, 2000), and the standard BH procedure applied to unscaled p-values (Vanilla). We conduct experiments on the WikiMIA benchmark using three LLMs and three detection scores (Perplexity, MIN-K%, and M-Entropy) at multiple target FIR levels. Table 1 presents that our method consistently achieves higher power across models, detection scores, and FIR targets. For example, under NeoX-20B with the MIN-K% score at $\alpha = 0.1$, our method improves power from 0.46% to 1.65%, a 258% relative gain over the vanilla baseline. Implementation details of the baselines are provided in Appendix E.2, and further discussion of related methods is in Appendix A.

**Robustness of error control to variations in $\pi_{\text{test}}$.** We assess the robustness of our method by evaluating its performance varying the proportion of training members in the test data, $\pi_{\text{test}}$. This analysis utilizes the MiniGPT-4 vision-language model (Zhu et al., 2024a) and the VL-MIA/Flickr dataset (Li et al., 2024c). The detection score $T(X)$ is based on the MaxRényi-K% (Li et al., 2024c), configured with hyperparameters $K = 100$ and $\gamma = 0.5$. The results presented in Figure 4 demonstrate that the achieved FIR is consistently bounded by the nominal level $\alpha$ across all tested values of $\pi_{\text{test}}$, thereby validating the effectiveness of our approach.

**Impact of calibration set size.** We investigate the effect of calibration set size on our method by evaluating FIR and power on ArXivTection with different models. Specifically, let $\rho = n/m$ denote the ratio of the calibration set size to the test set size. We vary $\rho$ in $\{0.1, 0.5, 1\}$. Figure 5

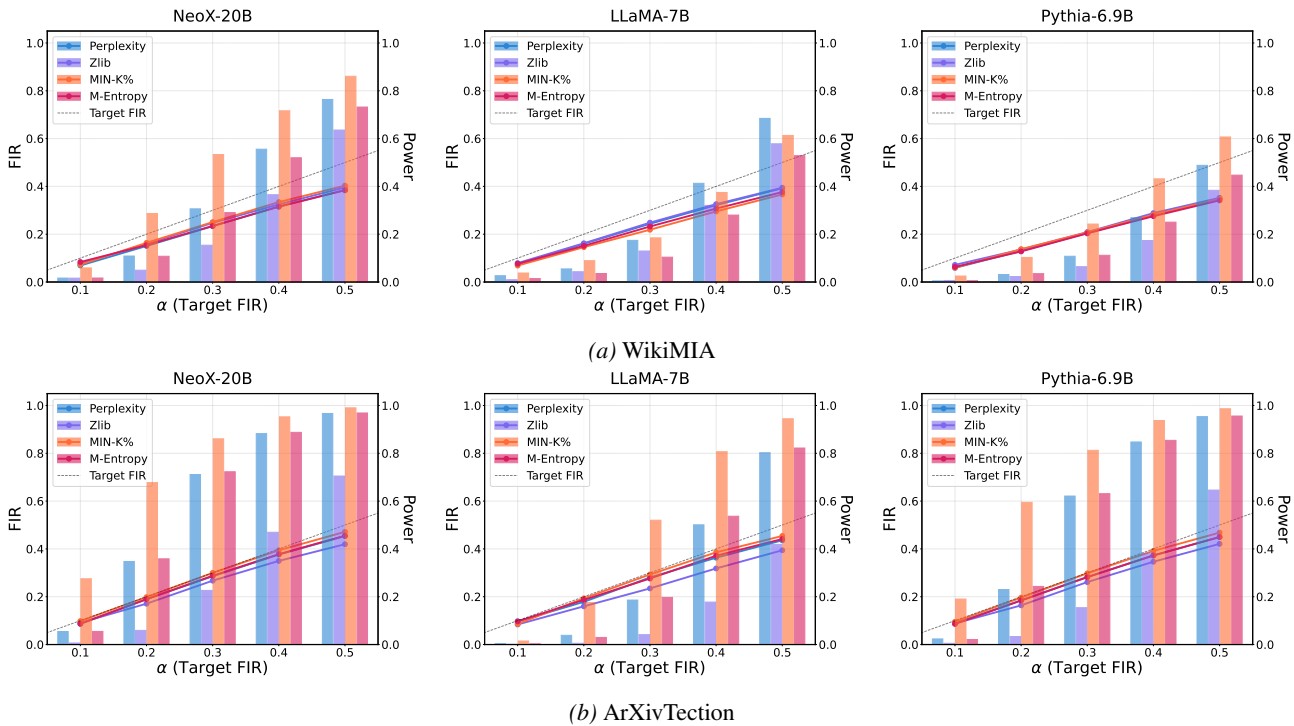

*(a)* WikiMIA

*(b)* ArXivTection

*Figure 1.* FIR (solid lines) and Power (bars) achieved by our method when applied to various detection scores across a range of levels $\alpha$ (target FIR). Each subplot corresponds to a specific model–dataset pair, and the dashed diagonal line indicates the target $\alpha$.

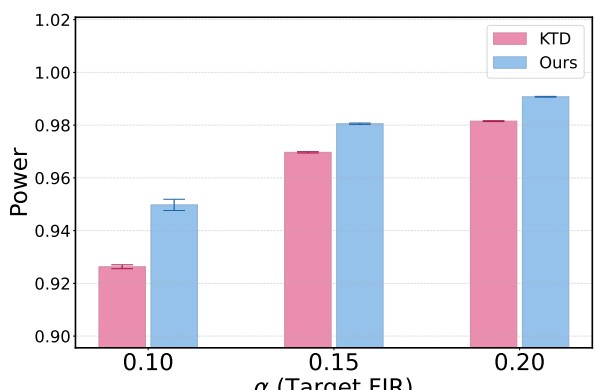

*Figure 2.* Power comparison with KTD on BBC (GPT-2). Error bars represent 95% confidence intervals

presents that our method effectively controls FIR across all tested $\rho$ values. In addition, increasing the calibration set size reduces the variance of FIP, resulting in more stable training data identification.

## 5. Discussion

**Leveraging confirmed training data to improve power.**
In some auditing scenarios, a calibration set containing a mix of confirmed **members** and **non-members** is available,

though with an arbitrary membership proportion. Such a set can be constructed by sampling from canonical public datasets known to be part of the model's training corpus (e.g., the Pile (Gao et al., 2020)) or by identifying instances of verbatim memorization. We argue that the information from known members can significantly enhance power. Accordingly, we propose a corresponding method based on the method of moments to estimate the data usage proportion.

For convenience, we define $\pi_0 = 1 - \pi_{\text{test}}$. The raw estimator by moment for $\pi_0$ is:

$$\hat{\pi}_{0,\text{raw}} = \frac{\hat{\mu}_1 - \hat{\mu}_{\text{test}}}{\hat{\mu}_1 - \hat{\mu}_0}$$

where $\hat{\mu}_0$, $\hat{\mu}_1$, and $\hat{\mu}_{\text{test}}$ are means of the detection scores derived from the non-member calibration set $\mathcal{D}^0_{\text{cal}}$, member calibration set $\mathcal{D}^1_{\text{cal}}$, and $\mathcal{D}_{\text{test}}$, respectively. To mitigate the positive bias in $1/\hat{\pi}_{0,\text{raw}}$ caused by Jensen's inequality, we introduce a bias-corrected estimator for the reciprocal $\theta_{1/\pi_0} = 1/\pi_0$, by subtracting an estimate of the leading bias term:

$$\hat{\theta}_{1/\pi_0} = \frac{1}{\hat{\pi}_{0,\text{raw}}} - \frac{\widehat{\text{Var}}(\hat{\pi}_{0,\text{raw}})}{\hat{\pi}^3_{0,\text{raw}}} \qquad (16)$$

To implement this, we approximate the variance $\text{Var}(\hat{\pi}_{0,\text{raw}})$

*Table 1.* Average power (%) comparison across different methods, LLMs, and detection scores at target FIR levels $\alpha$. Results with higher power are highlighted in **bold**, and ***Ours*** corresponds to the proposed method based on scaled p-values with the JKBB estimator. The $\pm$ values denote standard errors.

| Model | Method | $\alpha = 0.1$ | | | $\alpha = 0.2$ | | | $\alpha = 0.3$ | | |
|---|---|---|---|---|---|---|---|---|---|---|
| | | Perplexity | MIN-K% | M-Entropy | Perplexity | MIN-K% | M-Entropy | Perplexity | MIN-K% | M-Entropy |
| **NeoX-20B** | Vanilla | $0.46 \pm 0.03$ | $1.69 \pm 0.11$ | $0.48 \pm 0.04$ | $2.37 \pm 0.10$ | $7.69 \pm 0.24$ | $2.25 \pm 0.09$ | $6.77 \pm 0.22$ | $20.79 \pm 0.35$ | $7.31 \pm 0.24$ |
| | Storey-BH | $1.07 \pm 0.06$ | $3.54 \pm 0.17$ | $0.88 \pm 0.05$ | $6.00 \pm 0.22$ | $19.65 \pm 0.38$ | $5.24 \pm 0.21$ | $18.75 \pm 0.39$ | $42.81 \pm 0.38$ | $17.95 \pm 0.34$ |
| | BKY | $0.47 \pm 0.03$ | $1.76 \pm 0.12$ | $0.49 \pm 0.04$ | $2.46 \pm 0.11$ | $8.61 \pm 0.28$ | $2.40 \pm 0.10$ | $7.69 \pm 0.26$ | $27.25 \pm 0.47$ | $8.50 \pm 0.29$ |
| | Quantile-BH | $0.47 \pm 0.04$ | $1.73 \pm 0.11$ | $0.49 \pm 0.04$ | $2.39 \pm 0.10$ | $7.86 \pm 0.25$ | $2.30 \pm 0.10$ | $6.91 \pm 0.22$ | $21.17 \pm 0.36$ | $7.46 \pm 0.25$ |
| | ***Ours*** | $\mathbf{1.65 \pm 0.09}$ | $\mathbf{5.90 \pm 0.26}$ | $\mathbf{1.67 \pm 0.09}$ | $\mathbf{10.88 \pm 0.38}$ | $\mathbf{28.67 \pm 0.52}$ | $\mathbf{10.78 \pm 0.38}$ | $\mathbf{30.61 \pm 0.69}$ | $\mathbf{53.31 \pm 0.49}$ | $\mathbf{29.05 \pm 0.65}$ |
| **LLaMA-7B** | Vanilla | $0.90 \pm 0.04$ | $1.54 \pm 0.06$ | $0.46 \pm 0.03$ | $3.15 \pm 0.04$ | $4.82 \pm 0.08$ | $1.90 \pm 0.04$ | $3.86 \pm 0.05$ | $7.86 \pm 0.14$ | $2.72 \pm 0.04$ |
| | Storey-BH | $1.94 \pm 0.05$ | $3.16 \pm 0.08$ | $0.88 \pm 0.04$ | $3.61 \pm 0.05$ | $6.61 \pm 0.12$ | $2.39 \pm 0.04$ | $5.36 \pm 0.13$ | $12.31 \pm 0.19$ | $3.49 \pm 0.09$ |
| | BKY | $0.92 \pm 0.04$ | $1.58 \pm 0.07$ | $0.47 \pm 0.03$ | $3.18 \pm 0.04$ | $4.96 \pm 0.09$ | $1.91 \pm 0.04$ | $3.92 \pm 0.05$ | $8.47 \pm 0.16$ | $2.76 \pm 0.04$ |
| | Quantile-BH | $0.93 \pm 0.04$ | $1.60 \pm 0.06$ | $0.47 \pm 0.03$ | $3.16 \pm 0.04$ | $4.88 \pm 0.08$ | $1.92 \pm 0.04$ | $3.88 \pm 0.05$ | $8.04 \pm 0.14$ | $2.73 \pm 0.04$ |
| | ***Ours*** | $\mathbf{2.63 \pm 0.05}$ | $\mathbf{3.76 \pm 0.08}$ | $\mathbf{1.44 \pm 0.05}$ | $\mathbf{5.48 \pm 0.24}$ | $\mathbf{8.93 \pm 0.22}$ | $\mathbf{3.55 \pm 0.15}$ | $\mathbf{17.43 \pm 0.74}$ | $\mathbf{18.47 \pm 0.48}$ | $\mathbf{10.39 \pm 0.60}$ |
| **Pythia-6.9B** | Vanilla | $0.21 \pm 0.02$ | $0.88 \pm 0.06$ | $0.22 \pm 0.03$ | $1.04 \pm 0.06$ | $4.60 \pm 0.13$ | $1.11 \pm 0.08$ | $2.90 \pm 0.12$ | $10.05 \pm 0.21$ | $3.67 \pm 0.15$ |
| | Storey-BH | $0.41 \pm 0.04$ | $1.95 \pm 0.09$ | $0.40 \pm 0.04$ | $2.04 \pm 0.10$ | $7.34 \pm 0.18$ | $2.22 \pm 0.12$ | $6.28 \pm 0.21$ | $18.52 \pm 0.32$ | $6.84 \pm 0.20$ |
| | BKY | $0.21 \pm 0.02$ | $0.90 \pm 0.06$ | $0.22 \pm 0.03$ | $1.06 \pm 0.06$ | $4.82 \pm 0.14$ | $1.15 \pm 0.09$ | $3.03 \pm 0.13$ | $11.75 \pm 0.27$ | $3.88 \pm 0.16$ |
| | Quantile-BH | $0.22 \pm 0.02$ | $0.93 \pm 0.06$ | $0.22 \pm 0.03$ | $1.05 \pm 0.06$ | $4.67 \pm 0.13$ | $1.12 \pm 0.08$ | $2.92 \pm 0.12$ | $10.25 \pm 0.22$ | $3.70 \pm 0.15$ |
| | ***Ours*** | $\mathbf{0.47 \pm 0.04}$ | $\mathbf{2.49 \pm 0.11}$ | $\mathbf{0.55 \pm 0.05}$ | $\mathbf{3.13 \pm 0.14}$ | $\mathbf{10.28 \pm 0.27}$ | $\mathbf{3.49 \pm 0.17}$ | $\mathbf{10.78 \pm 0.38}$ | $\mathbf{24.24 \pm 0.44}$ | $\mathbf{11.18 \pm 0.38}$ |

*(a)* WikiMIA     *(b)* XSum     *(c)* BBC

*Figure 3.* Comparison of FIR control between our method and KTD across a range of levels $\alpha$ on three datasets

using the Delta method:

$$
\widehat{\mathrm{Var}}(\hat{\pi}_{0,\mathrm{raw}}) = \frac{1}{(\hat{\mu}_1 - \hat{\mu}_0)^2}\left[\hat{\sigma}_{0,\mathrm{raw}}^2 \frac{\hat{\sigma}_0^2}{n_0} \right.
$$
$$
\left. + (1 - \hat{\pi}_{0,\mathrm{raw}})^2 \frac{\hat{\sigma}_1^2}{n_1} + \frac{\hat{\sigma}_{\mathrm{test}}^2}{m}\right]. \tag{17}
$$

where $n_0 = |\mathcal{D}_{\mathrm{cal}}^0|$, $n_1 = |\mathcal{D}_{\mathrm{cal}}^1|$, and $m = |\mathcal{D}_{\mathrm{test}}|$. The terms $\hat{\sigma}_0^2, \hat{\sigma}_1^2, \hat{\sigma}_{\mathrm{test}}^2$ are the corresponding sample variances. The final estimator used in our algorithm is the adjusted moment estimator $\hat{\pi}_{\mathrm{mom}} = 1 - 1/\hat{\theta}_{1/\pi_0}$.

Crucially, for this extension to be reliable in legal or safety-critical contexts, it must maintain the rigorous error control of our original method. We now establish the theoretical guarantees, showing that $\hat{\pi}_{\mathrm{mom}}$ yields a consistent estimate of $\pi_{\mathrm{test}}$ and thereby preserves strict FIR control.

**Proposition 5.1.** *Assume the detection scores for member, non-member, and test distributions have finite first and second moments. As the sample sizes of the calibration and*

*test sets $n_0, n_1, m \to \infty$, the estimator $\hat{\pi}_{mom}$ is a consistent estimator for $\pi_{test}$. That is:*

$$
\hat{\pi}_{mom} \xrightarrow{p} \pi_{test},
$$

*where $\xrightarrow{p}$ denotes convergence in probability.*

This proposition shows that the adjusted moment estimator converges to the $\pi_{\mathrm{test}}$ as the data size is sufficiently large. We proceed by establishing the following theorem:

**Theorem 5.2.** *Under the conditions of Proposition 5.1, the PTDI procedure using the adjusted moment estimator $\hat{\pi}_{mom}$ achieves asymptotic FIR control. Specifically,:*

$$
\limsup_{n, m \to \infty} \mathrm{FIR} \leqslant \alpha.
$$

This theorem shows that the adjusted moment estimator strictly controls FIR. The corresponding proofs are provided in the Appendix B.3 and Appendix D.1. The details about the derivation of $\hat{\pi}_{\mathrm{mom}}$ is provided in Appendix D.

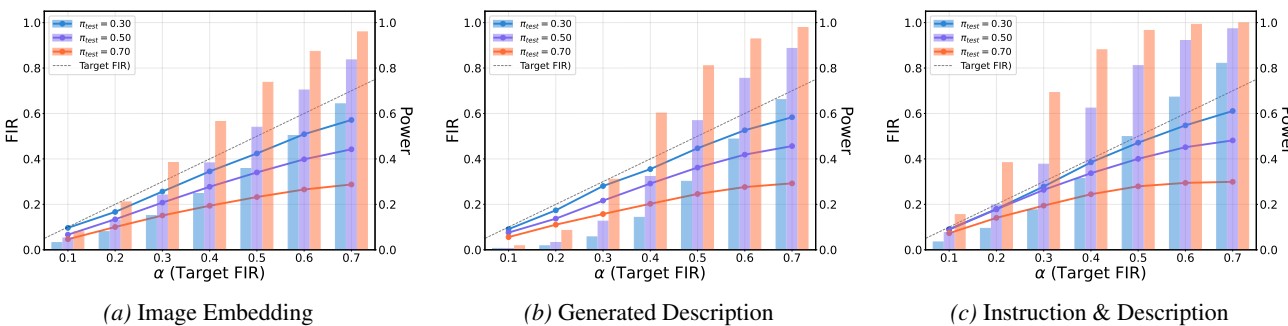

*Figure 4.* FIR (solid lines) and power (bars) achieved by our method on with the VL-MIA/Flickr dataset, evaluated across various data usage proportions of the test set $\pi_{\text{test}}$ and target FIR levels $\alpha$. All results are based on the MaxRényi-K% score calculated from three different input components: (a) the image embedding, (b) the generated description, and (c) the instruction combined with the description.

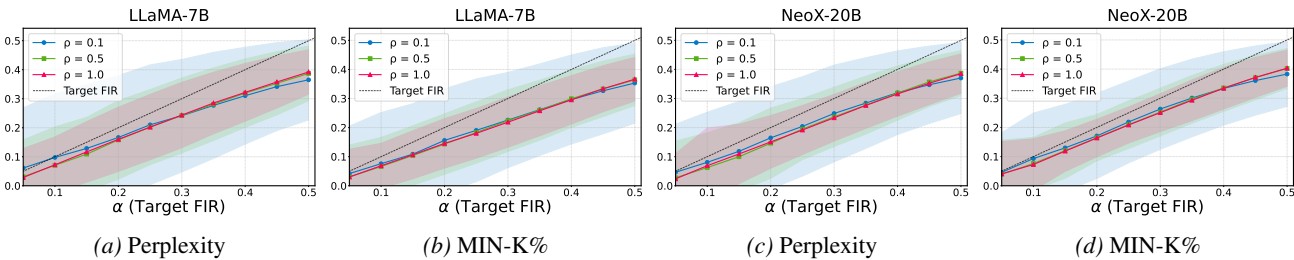

*Figure 5.* FIR curve achieved by our method under varying calibration set sizes. The parameter $\rho = n/m$ represents the ratio of the calibration set size ($n$) to the test set size ($m$). Shaded regions correspond to the mean ± one standard deviation.

The empirical results in Figure 6 align with the theoretical analysis, showing that the adjusted moment estimator can further improve power than the JKBB estimator while maintaining valid FIR control.

# 6. Related Work

**Training Data Detection in Large-Scale Models.** Identifying training data within large-scale models is a critical task with significant real-world implications, including ensuring fair model evaluation and providing credible evidence in copyright litigation. A primary concern in academic research is data contamination (Li, 2024), where benchmark data leaks into the training set, leading to untrustworthy evaluation results (Magar & Schwartz, 2022; Zhou et al., 2023; Li et al., 2024b). To address these issues, numerous studies have developed heuristic detection scores (Mattern et al., 2023; Xie et al., 2024; Zhang et al., 2024; Raoof et al., 2025; Yi & Li, 2026). These include metrics like perplexity (Carlini et al., 2022), MIN-k% (Shi et al., 2024), MIN-k%++ (Zhang et al., 2025c) for LLMs, and MaxRényi-K% for VLMs (Li et al., 2024c). However, these methods treat the task as a binary classification problem for individual points and lack the theoretical guarantees. *Additional discussion of MIA is deferred in Appendix A.*

Seeking to add statistical rigor, another line of work provides theoretical guarantees. For instance, Dekoninck et al.

(2024) uses multiple reference models to construct valid statistical tests, and Oren et al. (2023) leverages exchangeability for statistical inference. A key limitation, however, is that their guarantees apply only to dataset-level hypotheses—for example, determining if an entire dataset as a whole is contaminated. They are not designed for the fine-grained task of selecting a credible subset of individual data points. This is insufficient for practical applications, such as a copyright holder providing a specific list of infringed works, or an evaluator removing specific contaminated examples from a benchmark, a function supported by toolkits like lm-evaluation-harness (Gao et al., 2024). In this paper, we provide a method that determines a subset from a given dataset with provable error guarantees.

**Conformal inference.** Conformal inference provides distribution-free uncertainty quantification by exploiting exchangeability (Vovk et al., 2003; 2005). A central object in this framework is the conformal p-value, which compares the nonconformity score of a test point with those of calibration samples and is valid under the null hypothesis without requiring parametric assumptions. Classical conformal prediction (CP) uses these conformal p-values to construct a prediction set for a *single* test samples with marginal coverage guarantees (Lei & Wasserman, 2014; Angelopoulos & Bates, 2021; Huang et al., 2024; Xi et al., 2024; Liu et al., 2025a; Huang et al., 2025). By contrast, a recent line of work uses conformal p-values for selection and multiple test-

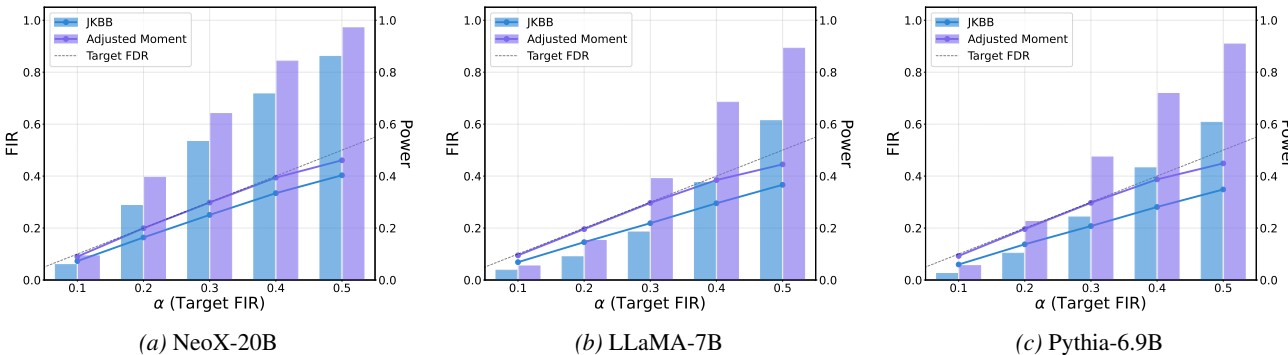

*Figure 6.* Performance of the JKBB and adjusted moment estimators on the WikiMIA dataset. Each plot shows the realized FIR (solid lines) and statistical power (bars) for a given model.

ing, where the goal is to select a subset of instances while controlling a set-level error criterion such as the false discovery rate; this paradigm is often referred to as conformal selection (Bates et al., 2023; Jin & Candès, 2023; Bai et al., 2025; Jin & Candès, 2026; Hao et al., 2026). Following this latter direction, we use conformal p-values to cast training data identification as a multiple-testing problem and control the false identification rate of the selected set. *A more detailed discussion on controlling the false identification rate is provided in Appendix A.*

**Estimating Data Usage Proportion.** A key component of our method's ability to improve power is the estimation of the data usage proportion. This problem was recently formalized as Dataset Usage Cardinality Inference (DUCI) by Tong et al. (2025). However, this approach requires training reference models to estimate the necessary statistics, rendering it unsuitable for pre-training data detection in large-scale models where the training process is opaque and prohibitively expensive. In contrast, our proposed JKBB estimator is significantly more practical, as it only requires access to the target model, a set of confirmed non-member data, and the test set, making it a more versatile tool for real-world auditing scenarios.

## 7. Conclusion

In this paper, we formalize training-data identification as a set-level inference problem and introduce PTDI, a method that provides rigorous statistical guarantees for the identified set. Our method leverages conformal p-values and the BH procedure to obtain distribution-free guarantees using only non-training calibration data. To improve power, we introduce a novel JKBB estimator for the data usage proportion. Leveraging this estimator, we apply a p-value scaling strategy that enhances power while maintaining theoretical guarantees. When a small set of confirmed training data is available, we introduce an adjusted moment estimator to further improve the power. Extensive experiments show that

PTDI consistently outperforms existing methods in power while maintaining strict FIR control across diverse detection scores. Our method enables reliable training data identification in high-stakes applications, such as data copyright litigation, where strict false identification control is required.

**Limitation and further work.** Though our method provides rigorous theoretical guarantees to control the false identification, it requires a calibration set of unseen data that is distributionally similar to the test set. A substantial divergence between the calibration and test data may compromise the validity of these guarantees. To address this issue, handling distribution shift by weighted conformal p-values may be a promising direction for future work.

**Acknowledgements** This research is supported by Guangdong Basic and Applied Basic Research Foundation (Grant No. 2026A1515011367). This project is also supported by the Jiangsu Provincial Key Discipline Construction Project (Statistics) and the open project of the Joint Lab for Statistics and Finance (Grant No. 2025JLSF101). This research is supported by the SUSTech-NUS Joint Research Program. Weiran Huang is supported by the National Natural Science Foundation of China (No. 62406192), Shanghai Municipal Special Program for Basic Research on General AI Foundation Models (Grant No. 2025SHZDZX025G03), Tencent WeChat Rhino-Bird Focused Research Program, and Kuaishou Technology. We gratefully acknowledge the support of the Center for Computational Science and Engineering at the Southern University of Science and Technology for our research.

## Impact Statement

This paper presents work whose goal is to advance the field of Machine Learning. There are many potential societal consequences of our work, none of which we feel must be specifically highlighted here.

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

# A. Additional Related Work

**Membership Inference Attacks.**   From a privacy perspective, Membership Inference Attacks (MIAs) aim to determine if a specific data point was used to train a target model, which could expose sensitive information (Shokri et al., 2017; Yeom et al., 2018; Salem et al., 2019). A significant body of work treats this as a binary classification problem, relying on scores computed without reference models, such as loss (Yeom et al., 2018), entropy (Yeom et al., 2018), confidence (Liu et al., 2019) and gradient norm (Nasr et al., 2019; Sablayrolles et al., 2019). While accessible, this approach focuses on average classification accuracy. Shifting to a more statistically-minded viewpoint, many prominent works correctly frame MIA as a hypothesis test, including Attack-P (Ye et al., 2022) and other prominent works (Carlini et al., 2022; Zarifzadeh et al., 2024). These approaches prioritize metrics like the true positive rate (TPR) at a low false positive rate (FPR), but this still relies on an average-case error metric and fails to provide a formal statistical guarantee for any individual inference. Furthermore, some attacks like Attack-R (Ye et al., 2022) and BMIA (Liu et al., 2025b) estimate the conditional distribution of individual data points to control type-I error for specific inferences, but they require at least one reference model, making them unsuitable for detecting pre-training data in LLMs. Moreover, the average type-I error (FPR) is ill-suited for multiple-hypothesis testing, where controlling the FIR is more appropriate for ensuring credible evidence across the selected membership set, as discussed in Appendix G. In this work, we propose a versatile method that ensures strict FIR control and can be integrated with most MIA methods.

**False Identification Rate Control.**   In this paper, we define the false identification rate as the key metric for provable training data identification. This notion is inspired by the false discovery rate (FDR) (Benjamini & Hochberg, 1995), which was originally introduced to control false discoveries when rejecting null hypotheses in favor of new findings, such as an effective treatment. Analogously, we introduce the False Identification Rate (FIR) to better align with the objective of training data identification, where the goal is to identify a set of training samples with a controlled risk of false identification. A standard approach for controlling such set-level error is the Benjamini–Hochberg (BH) procedure (Benjamini & Hochberg, 1995; Benjamini & Yekutieli, 2001), but it is conservative since it assumes that all null hypotheses might be true (number of true nulls $m_0 = m$). To address this, a variety of adaptive methods including BH-Storey (Storey, 2002), BKY (Benjamini et al., 2006), and Quantile-BH (Benjamini & Hochberg, 2000) are proposed to estimate the true proportion of null hypotheses $\pi_0$. However, the most widely used BH-Storey estimator is intrinsically biased as it estimates the p-value density at the boundary ($p = 1$) by averaging over a fixed interval. In settings where the p-value density exhibits a downward trend near the boundary, this method yields an inflated estimate of $\pi_0$, leading to a conservative procedure that compromises statistical power (Neuvial, 2013). In this paper, we propose a novel JKBB estimator with provable properties and a task-specific optimal bandwidth, which achieves higher power than existing adaptive BH procedures.

Regarding the specific task of training data detection, Hu et al. (2025) proposed a method based on knockoff statistics to control FIR. However, its effectiveness hinges on generating high-quality knockoffs (distributed symmetrically), a challenging task that can lead to unstable FIR control in practice. Moreover, their approach does not explicitly attribute the failure of membership inference to provide reliable evidence in large-scale models (e.g., LLMs and VLMs) to the limitations of instance-wise hypothesis testing. In this paper, our work builds on the distribution-free conformal inference framework (Vovk et al., 2003; 2005; Bates et al., 2023; Jin & Candès, 2023), allowing us to achieve rigorous FIR control without relying on strong distributional assumptions.

**Estimating Data Usage Proportion.**   A key component of our method's ability to improve power is the estimation of the data usage proportion. This problem was recently formalized as Dataset Usage Cardinality Inference (DUCI) by Tong et al. (2025). However, this approach requires training reference models to estimate the necessary statistics, rendering it unsuitable for pre-training data detection in large-scale models where the training process is opaque and prohibitively expensive. In contrast, our proposed JKBB estimator is significantly more practical, as it only requires access to the target model, a set of confirmed non-member data, and the test set, making it a more versatile tool for real-world auditing scenarios.

# B. Proofs

## B.1. Auxiliary Lemmas and Propositions

**Lemma B.1** (Classical FIR Control under PRDS (Benjamini & Yekutieli, 2001)). *Given a set of p-values $\{p_j\}_{j=1}^m$ from $m$ hypothesis tests, of which $m_0$ are true null hypotheses. If the p-values corresponding to the true nulls are valid (i.e., they are super-uniformly distributed under the null), and if the entire p-value vector satisfies the Positive Regression Dependency on*

*Table 2.* Summary of notations used in the paper.

| Notation | Description |
|---|---|
| $\theta$ | The target model. |
| $\mathcal{D}_{\text{train}}$ | The set of data used to train the model $\theta$. |
| $X, X_i$ | A data point and the $i$-th data point, respectively. |
| $M, M_i$ | true membership label for a data point $X$, $X_i$ (1 if it is in traning set $\mathcal{D}_{\text{train}}$, otherwise 0). |
| $\mathcal{D}_{\text{cal}}, \mathcal{D}_{\text{test}}$ | The calibration set and the test set, respectively. |
| $n, m$ | The number of samples in the calibration and test sets, respectively. |
| $T(X; \theta)$ | The detection score for data point $X$ (e.g., perplexity). Lower is more member-like. |
| $H_{0,j}$ | The null hypothesis that test point $X_{n+j}$ is a non-member ($M_{n+j} = 0$). |
| $\mathcal{S}$ | The selected subset of indices from $\mathcal{D}_{\text{test}}$ rejected as null hypotheses. |
| FIP | The proportion of false identification in $\mathcal{S}$. |
| FIR | The expected proportion of false identification in $\mathcal{S}$. |
| Power | The expected proportion of all true members that are correctly identified in $\mathcal{S}$. |
| $\alpha$ | The target False Discovery Rate (FIR) level. |
| $\pi_{\text{test}}$ | The true proportion of training members in the test set. |
| $\hat{\pi}_{\text{test}}$ | An estimate of the data usage proportion $\pi_{\text{test}}$. |
| $p_j$ | The initial conformal p-value for the $j$-th test point. |
| $\tilde{p}_j$ | The scaled p-value, calculated as $(1 - \hat{\pi}_{\text{test}})p_j$. |
| $\mathcal{E}$ | A generic estimator function for the data usage proportion. |
| $\hat{\pi}_{\text{bdy}}$ | The Jackknife-corrected Beta Boundary (JKBB) estimator. |
| $f_{\text{test}}(p)$ | The marginal density function of the test p-values on $[0, 1]$. |
| $b$ | Bandwidth parameter for the Beta kernel. |
| $\gamma$ | Scaling factor for the Jackknife dual-bandwidth construction. |
| $w_0, w_1$ | Weights for the Jackknife combination ($w_0 = \frac{\gamma}{\gamma-1}, w_1 = -\frac{1}{\gamma-1}$). |
| $K_{1,b}(t)$ | The boundary Beta kernel function evaluated at $p = 1$. |
| $c_1, c_2$ | First and second-order bias coefficients of the density estimator. |
| $\Omega(\gamma)$ | The variance constant depending on the step size $\gamma$. |
| $\hat{\pi}_{\text{mom}}$ | The adjusted moment-based estimator for $\pi_{\text{test}}$. |
| $\eta$ | A quantile hyperparameter used to define the region $\mathcal{R}$. |
| $\mu_0, \mu_1$ | The true mean of detection scores for non-members and members. |
| $\hat{\mu}_0, \hat{\mu}_1$ | The sample mean of detection scores for non-members and members. |
| $m_0, m_1$ | The number of non-member and member samples in the test set, respectively. |

*a Subset (PRDS) property, then the standard Benjamini-Hochberg (BH) procedure at a target level $\alpha$ controls the False Discovery Rate (FIR) such that:*

$$\text{FIR} \leqslant \frac{m_0}{m}\alpha.$$

**Proposition B.2** (FIR Control for the BH Procedure on $p_j$)**.** *Let the p-values $\{p_j\}_{j=1}^m$ be constructed as defined in Equation* (6)*. Assume that each sample's membership status in the test set $\mathcal{D}_{\text{test}}$ is an independent Bernoulli trial, where the probability of being a member ($M = 1$) is $\pi_{\text{test}}$. Further assume that the p-value vector $(p_1, \ldots, p_m)$ satisfies the PRDS property.*

*If the standard Benjamini-Hochberg (BH) procedure is applied directly to these p-values $\{p_j^0\}$ at a target level $\alpha$, then its FIR is controlled such that:*

$$\text{FIR} \leqslant \alpha \cdot (1 - \pi_{\text{test}}).$$

*Proof.* First, we verify that the p-values $\{p_j\}$ generated by Equation (6) satisfy the preconditions of Lemma B.1.

For any true null hypothesis $H_{0,j}$ (i.e., $M_{n+j} = 0$), the p-value $p_j$ is constructed using a calibration set composed entirely of non-members. According to the principles of conformal prediction, when the test sample $X_{n+j}$ is also a non-member, its score $T_{n+j}$ is exchangeable with the scores from the calibration set. This construction guarantees that $p_j$ is super-uniformly distributed under the null hypothesis, meaning $P(p_j \leqslant t \mid j \in \mathcal{H}_0) \leqslant t$ for all $t \in [0, 1]$.

The PRDS property is validated by Lemma B.1 of Bates et al. (2023); Jin & Candès (2023). Since both conditions are met, we can apply the conclusion of Lemma B.1 for a fixed $\mathcal{D}_{\text{test}}$:

$$\text{FIR} = \mathbb{E}[\text{FIP} \mid \mathcal{D}_{\text{test}}] \leqslant \frac{m_0}{m}\alpha$$

Recall that $\pi_{\text{test}} = \Pr(M_j = 1)$. For a fixed test set $\mathcal{D}_{\text{test}}$, this probability corresponds to the actual proportion of members, i.e., $\pi_{\text{test}} = 1 - m_0/m$. By substituting, we obtain:

$$\text{FIR} \leqslant (1 - \pi_{\text{test}}) \cdot \alpha.$$

$\square$

**Proposition B.3** (Asymptotic FIR control of the plug-in scaled BH procedure). *Let $\{p_j\}_{j=1}^m$ denote the original p-values, and consider the PTDI procedure that applies the Benjamini–Hochberg (BH) procedure at nominal level $\alpha$ to the scaled p-values*

$$\tilde{p}_j = \hat{\pi}_{0,m}\, p_j, \qquad j = 1, \ldots, m,$$

*where $\hat{\pi}_{0,m} := 1 - \hat{\pi}$ is an estimator of the null proportion $\pi_{0,\text{test}} = 1 - \pi_{\text{test}}$. Let $\text{FIR}_m$ denote the false identification rate of the resulting rejection set. Then the plug-in procedure satisfies the asymptotic FIR bound*

$$\limsup_{m \to \infty} \text{FIR}_m \leqslant \frac{1 - \pi_{\text{test}}}{\pi_{0,\infty}}\alpha.$$

*Proof.* Let $p_{(1)} \leqslant \cdots \leqslant p_{(m)}$ be the ordered p-values. As shown, the plug-in procedure is equivalent to the standard BH procedure applied at a random level $\alpha'_m = \alpha/\hat{\pi}_{0,m}$.

We denote $\text{FIP}(\alpha)$ as the FIP of the BH procedure at level $\alpha$ for a fixed set of p-values. A key property of the BH procedure is that the rejection set is nested and the FIP is asymptotically non-decreasing with respect to the significance level $\alpha$.

By the assumption $\hat{\pi}_{0,m} \xrightarrow{P} \pi_{0,\infty}$, for any small $\epsilon > 0$, the event $E_m = \{\alpha'_m \leqslant \alpha/\pi_{0,\infty} + \epsilon\}$ occurs with probability tending to 1 as $m \to \infty$. On the event $E_m$, by the monotonicity of the BH procedure:

$$\text{FIP}(\alpha'_m) \leqslant \text{FIP}\left(\frac{\alpha}{\pi_{0,\infty}} + \epsilon\right).$$

Taking expectations and applying the result from Proposition B.2 (which holds for any fixed level), we have:

$$\mathbb{E}[\text{FIP}(\alpha'_m)\mathbb{I}_{E_m}] \leq \mathbb{E}\left[\text{FIP}\left(\frac{\alpha}{\pi_{0,\infty}} + \epsilon\right)\right] \leq (1 - \pi_{\text{test}})\left(\frac{\alpha}{\pi_{0,\infty}} + \epsilon\right).$$

As $m \to \infty$ and $\epsilon \to 0$, since the FIP is bounded in $[0, 1]$, the contribution of the complement event $E_m^c$ vanishes. Therefore:

$$\limsup_{m \to \infty} \text{FIR}_m \leqslant \frac{1 - \pi_{\text{test}}}{\pi_{0,\infty}}\alpha.$$

This completes the proof. $\square$

## B.2. The proof of Theorem 3.1

*Proof.* The proof relies on the consistency of the boundary estimator and the asymptotic FIR bound established in Proposition B.3.

As we will formally prove in Proposition C.1, $\hat{\pi}_{0,m}$ converges in probability to the marginal probability density of the test p-values evaluated at 1:

$$\hat{\pi}_{0,m} = 1 - \hat{\pi}_{\text{bdy}} \xrightarrow{P} f_{\text{test}}(1) \quad \text{as } m \to \infty. \tag{18}$$

Let $\pi_{0,\infty} := f_{\text{test}}(1)$. According to Proposition B.3, the asymptotic FIR of the plug-in procedure satisfies:

$$\limsup_{m \to \infty} \text{FIR}_m \leqslant \frac{1 - \pi_{\text{test}}}{\pi_{0,\infty}} \alpha. \tag{19}$$

Since the density of member p-values is non-negative, i.e., $f(1 \mid M = 1) \geqslant 0$, the limit $\pi_{0,\infty} := f_{\text{test}}(1)$ serves as a conservative upper bound for the true null proportion $1 - \pi_{\text{test}}$:

$$\pi_{0,\infty} = (1 - \pi_{\text{test}}) + \pi_{\text{test}} f(1 \mid M = 1) \geqslant 1 - \pi_{\text{test}}. \tag{20}$$

Hence, we conclude that

$$\limsup_{m \to \infty} \text{FIR}_m \leqslant \frac{1 - \pi_{\text{test}}}{1 - \pi_{\text{test}}} \alpha = \alpha.$$

This completes the proof. □

## B.3. The Proof of Proposition 5.1

*Proof.* We prove the consistency of $\hat{\pi}_{\text{mom}}$ by relying on the Weak Law of Large Numbers (WLLN) and the Continuous Mapping Theorem (CMT). Let $\mu_0, \mu_1$ and $\sigma_0^2, \sigma_1^2$ be the true means and variances of the detection scores for the non-member and member populations, respectively. The true mean of the test set is $\mu_{\text{test}} = (1 - \pi_{\text{test}})\mu_0 + \pi_{\text{test}}\mu_1$, and we define the true non-member proportion as $\pi_0 = 1 - \pi_{\text{test}}$.

Under the assumption of finite moments, the WLLN ensures that the sample means $\hat{\mu}_0, \hat{\mu}_1, \hat{\mu}_{\text{test}}$ and sample variances $\hat{\sigma}_0^2, \hat{\sigma}_1^2, \hat{\sigma}_{\text{test}}^2$ converge in probability to their true population counterparts. Since the raw estimator $\hat{\pi}_{0,\text{raw}}$ is a continuous function of these sample means, the CMT implies its convergence in probability. Specifically, assuming $\mu_1 \neq \mu_0$ for the score to be informative, we have

$$\hat{\pi}_{0,\text{raw}} = \frac{\hat{\mu}_1 - \hat{\mu}_{\text{test}}}{\hat{\mu}_1 - \hat{\mu}_0} \xrightarrow{p} \frac{\mu_1 - \mu_{\text{test}}}{\mu_1 - \mu_0} = \frac{\mu_1 - ((1 - \pi_{\text{test}})\mu_0 + \pi_{\text{test}}\mu_1)}{\mu_1 - \mu_0} = \pi_0.$$

Next, we consider the bias-correction term $\frac{\widehat{\text{Var}}(\hat{\pi}_{0,\text{raw}})}{\hat{\pi}_{0,\text{raw}}^3}$. Recall that

$$\widehat{\text{Var}}(\hat{\pi}_{0,\text{raw}}) = \frac{1}{(\hat{\mu}_1 - \hat{\mu}_0)^2} \left[ \hat{\pi}_{0,\text{raw}}^2 \frac{\hat{\sigma}_0^2}{n_0} + (1 - \hat{\pi}_{0,\text{raw}})^2 \frac{\hat{\sigma}_1^2}{n_1} + \frac{\hat{\sigma}_{\text{test}}^2}{m} \right],$$

where $n_0 = |\mathcal{D}_{\text{cal}}^0|$, $n_1 = |\mathcal{D}_{\text{cal}}^1|$, and $m = |\mathcal{D}_{\text{test}}|$. As $n_0, n_1, m \to \infty$, the estimated variance $\widehat{\text{Var}}(\hat{\pi}_{0,\text{raw}})$ converges in probability to zero because its constituent terms are scaled by $1/n_0, 1/n_1$, or $1/m$, and the $\hat{\pi}_{0,\text{raw}}^3$ converges in probability to the non-zero constant $\pi_0^3$. Thus, the entire correction term converges to zero by Slutsky's theorem. It follows that $\hat{\theta}_{1/\pi_0} = 1/\hat{\pi}_{0,\text{raw}} - \frac{\widehat{\text{Var}}(\hat{\pi}_{0,\text{raw}})}{\hat{\pi}_{0,\text{raw}}^3} \xrightarrow{P} 1/\pi_0$. Finally, since the final estimator is a continuous transformation of $\hat{\theta}_{1/\pi_0}$, another application of the CMT yields the desired result:

$$\hat{\pi}_{\text{mom}} = 1 - \frac{1}{\hat{\theta}_{1/\pi_0}} \xrightarrow{p} 1 - \frac{1}{1/\pi_0} = 1 - \pi_0 = \pi_{\text{test}}.$$

This completes the proof. □

### B.4. Scaling procedure improves Power

**Theorem B.4.** *Let $\mathcal{S}_0$ and $\mathcal{S}_1$ be the selection sets from the standard Benjamini-Hochberg procedure and the scaling procedure, respectively. If $0 < \hat{\pi}_{test} < 1$, then we have:*

$$Power(\mathcal{S}_1) \geqslant Power(\mathcal{S}_0)$$

*Proof.* Let $p_{(1)} \leqslant \cdots \leqslant p_{(m)}$ be the sorted p-values. Suppose that selection sets are determined by the stopping indices $k_0$ and $k_1$:

$$\mathcal{S}_0 = \{j \mid p_j \leqslant p_{(k_0)}\}, \ \mathcal{S}_1 = \{j \mid p_j \leqslant p_{(k_1)}\},$$

where $k_0 = \max\left\{k \mid p_{(k)} \leqslant \frac{k}{m}\alpha\right\}$ and $k_1 = \max\left\{k \mid \tilde{p}_{(k)} \leqslant \frac{k}{m}\alpha\right\}$.

Substituting $\tilde{p}_{(k)} = (1 - \hat{\pi}_{\text{test}})p_{(k)}$ into the condition for $k_1$ yields the equivalent inequality:

$$\tilde{p}_{(k)} \leqslant \frac{k}{m}\alpha \iff (1 - \hat{\pi}_{\text{test}})p_{(k)} \leqslant \frac{k}{m}\alpha \iff p_{(k)} \leqslant \frac{k}{m}\left(\frac{\alpha}{1 - \hat{\pi}_{\text{test}}}\right)$$

Since $0 < \hat{\pi}_{\text{test}} < 1$, strictly weaker rejection criterion applies to $\mathcal{S}_1$:

$$p_{(k)} \leqslant \frac{k}{m}\alpha \implies p_{(k)} \leqslant \frac{k}{m}\alpha \leqslant \frac{k}{m}\left(\frac{\alpha}{1 - \hat{\pi}_{\text{test}}}\right)$$

This implication ensures that any index $k$ satisfying the condition for $\mathcal{S}_0$ also satisfies it for $\mathcal{S}_1$. Therefore:

$$k_0 \leqslant k_1 \implies p_{(k_0)} \leqslant p_{(k_1)} \implies \mathcal{S}_0 \subseteq \mathcal{S}_1$$

By the definition of power, we have:

$$Power(\mathcal{S}_1) \geqslant Power(\mathcal{S}_0) \tag{21}$$

This completes the proof.

$\square$

## C. Detail Derivation for JKBB Estimator

In this section, we derive the optimal bandwidth choice for the Jackknife-corrected Beta Boundary (JKBB) estimator at the boundary point $p = 1$, and establish its consistency under standard regularity conditions. We start by deriving the bias.

### C.1. Bias Expansion of the Boundary Beta Kernel Estimator

Let $f$ be a probability density supported on $[0, 1]$, twice continuously differentiable in a neighborhood of $p = 1$. The Beta boundary kernel estimator at $p = 1$ with bandwidth $b > 0$ is defined as

$$K_{1,b}(t) = \left(\frac{1}{b} + 1\right) t^{\frac{1}{b}}, \qquad t \in [0, 1]. \tag{22}$$

Given i.i.d. samples $\{p_j\}_{j=1}^m$ drawn from $f$, the estimator is

$$\widehat{f}_b(1) = \frac{1}{m}\sum_{j=1}^m K_{1,b}(p_j).$$

Its expectation admits the integral representation

$$\mathbb{E}[\widehat{f}_b(1)] = \int_0^1 \left(\frac{1}{b} + 1\right) t^{\frac{1}{b}} f(t) \, dt. \tag{23}$$

Since the kernel mass concentrates near $t = 1$ as $b \to 0$, we expand $f(t)$ around $t = 1$:

$$f(t) = f(1) - f'(1)(1 - t) + \frac{1}{2}f''(1)(1 - t)^2 + O((1 - t)^3).$$

Substituting into Equation (23) yields

$$\mathbb{E}[\widehat{f_b}(1)] = f(1)I_0 - f'(1)I_1 + \frac{1}{2}f''(1)I_2 + O(b^3),$$

where the kernel moments are

$$I_k = \int_0^1 (1-t)^k \frac{1}{b} t^{\frac{1}{b}-1} \, dt = \frac{1}{b}B\left(\frac{1}{b}, k+1\right).$$

Using the identity

$$\int_0^1 t^{a-1}(1-t)^k dt = B(a, k+1) = \frac{\Gamma(a)\Gamma(k+1)}{\Gamma(a+k+1)},$$

and the Gamma recursion formula, the kernel moments admit the closed form

$$I_k = \frac{1}{b}\frac{k!}{(\frac{1}{b})(\frac{1}{b}+1)\cdots(\frac{1}{b}+k)}.$$

A direct expansion in $b$ then yields

$$I_0 = 1, \qquad I_1 = b - 2b^2 + O(b^3), \qquad I_2 = 2b^2 + O(b^3).$$

Combining terms, the bias expansion becomes

$$\mathbb{E}[\widehat{f_b}(1)] = f(1) - f'(1)b + \big(2f'(1) + f''(1)\big)b^2 + O(b^3). \tag{24}$$

Thus, the first and second-order bias coefficients are

$$c_1 = -f'(1), \qquad c_2 = 2f'(1) + f''(1).$$

## C.2. Jackknife Bias Correction

To eliminate the leading $O(b)$ bias term in Equation (24), we adopt a dual-kernel jackknife construction. For a fixed $\gamma > 1$, define

$$\widehat{f}_{\mathrm{jk}}(1) = w_0 \widehat{f_b}(1) + w_1 \widehat{f}_{\gamma b}(1), \tag{25}$$

where the weights satisfy

$$w_0 + w_1 = 1, \qquad w_0 + \gamma w_1 = 0,$$

which yields

$$w_0 = \frac{\gamma}{\gamma - 1}, \quad w_1 = -\frac{1}{\gamma - 1}.$$

By linearity of expectation and the expansion

$$\mathbb{E}[\widehat{f}_h(1)] = f(1) + c_1 h + c_2 h^2 + O(h^3),$$

we have

$$\mathbb{E}[\widehat{f}_{\mathrm{jk}}(1)] = f(1) + c_1(w_0 + \gamma w_1)b + c_2(w_0 + \gamma^2 w_1)b^2 + O(b^3)$$
$$= f(1) - \gamma c_2 b^2 + O(b^3).$$

Hence, the jackknife-corrected estimator has bias of order $O(b^2)$:

$$\mathrm{Bias}[\widehat{f}_{\mathrm{jk}}(1)] = -\gamma c_2 b^2 + O(b^3). \tag{26}$$

## C.3. Variance and Mean Squared Error Analysis

Recall the definition of the boundary Beta kernel

$$K_{1,b}(t) = \left(\frac{1}{b} + 1\right) t^{1/b}, \qquad t \in [0, 1], \tag{27}$$

and the associated boundary density estimator

$$\widehat{f}_b(1) = \frac{1}{m} \sum_{j=1}^{m} K_{1,b}(P_j). \tag{28}$$

We first verify the normalization property. Direct integration yields

$$\int_0^1 K_{1,b}(t)\, dt = \left(\frac{1}{b} + 1\right) \int_0^1 t^{1/b}\, dt = \left(\frac{1}{b} + 1\right) \frac{1}{1/b + 1} = 1.$$

Next, we analyze the $L^2$ norm of the kernel. A direct calculation yields

$$\begin{aligned}
\int_0^1 K_{1,b}(t)^2\, dt &= \left(\frac{1}{b} + 1\right)^2 \int_0^1 t^{2/b}\, dt \\
&= \frac{(1+b)^2}{b^2} \cdot \frac{1}{2/b + 1} \\
&= \frac{(1+b)^2}{b(2+b)} \\
&= 1 + \frac{1}{2b + b^2} \\
&= 1 + \frac{1}{2b(1 + b/2)}.
\end{aligned} \tag{29}$$

Using the expansion $(1 + b/2)^{-1} = 1 - b/2 + O(b^2)$ as $b \to 0$, we verify the asymptotic behavior:

$$\begin{aligned}
\int_0^1 K_{1,b}(t)^2\, dt &= 1 + \frac{1}{2b}\left(1 - b/2 + O(b^2)\right) \\
&= \frac{1}{2b} + O(1).
\end{aligned} \tag{30}$$

We now derive the variance of the Jackknife estimator $\widehat{f}_{\mathrm{jk}}(1) = w_0 \widehat{f}_b(1) + w_1 \widehat{f}_{\gamma b}(1)$. To this end, we first compute the cross-term inner product between the two boundary kernels. A direct calculation yields

$$\begin{aligned}
\int_0^1 K_{1,b}(t) K_{1,\gamma b}(t)\, dt &= \left(\frac{1}{b} + 1\right)\left(\frac{1}{\gamma b} + 1\right) \int_0^1 t^{\frac{1}{b} + \frac{1}{\gamma b}}\, dt \\
&= \left(\frac{1}{b} + 1\right)\left(\frac{1}{\gamma b} + 1\right) \frac{1}{\frac{1}{b} + \frac{1}{\gamma b} + 1} \\
&= \frac{(1+b)(1+\gamma b)}{b(\gamma + 1 + \gamma b)} \\
&= \frac{1}{b(\gamma + 1)} \cdot \frac{(1+b)(1+\gamma b)}{1 + \frac{\gamma}{\gamma+1} b} \\
&= \frac{1}{b(\gamma + 1)} \left(1 + (\gamma + 1)b + \gamma b^2\right)\left(1 - \frac{\gamma}{\gamma + 1} b + O(b^2)\right) \\
&= \frac{1}{b(\gamma + 1)} \left(1 + \frac{\gamma^2 + \gamma + 1}{\gamma + 1} b + O(b^2)\right) \\
&= \frac{1}{b(\gamma + 1)} + \frac{\gamma^2 + \gamma + 1}{(\gamma + 1)^2} + O(b) = \frac{1}{b(\gamma + 1)} + O(1).
\end{aligned} \tag{31}$$

Defining the composite kernel $Z(p) = w_0 K_{1,b}(p) + w_1 K_{1,\gamma b}(p)$, the variance is given by $\frac{1}{m}(\mathbb{E}[Z^2] - \mathbb{E}[Z]^2)$. Using the approximations Equation (30) and Equation (31), and approximating $f(t) \approx f(1)$ near the boundary, we obtain:

$$
\begin{aligned}
\mathrm{Var}\big(\widehat{f}_{\mathrm{jk}}(1)\big) &= \frac{1}{m}\Big(\mathbb{E}[Z^2] - \mathbb{E}[Z]^2\Big) \\
&\approx \frac{1}{m}\mathbb{E}[Z^2] && (32) \\
&\quad \text{(since } \mathbb{E}[Z]^2 = O(1) \ll \mathbb{E}[Z^2]) \\
&= \frac{1}{m}\int_0^1 \Big(w_0 K_{1,b}(t) + w_1 K_{1,\gamma b}(t)\Big)^2 f(t)\, dt \\
&\approx \frac{f(1)}{m}\int_0^1 \Big(w_0 K_{1,b}(t) + w_1 K_{1,\gamma b}(t)\Big)^2 dt && (33) \\
&= \frac{f(1)}{m}\Bigg[ w_0^2 \underbrace{\|K_{1,b}\|_2^2}_{\approx \frac{1}{2b}} + w_1^2 \underbrace{\|K_{1,\gamma b}\|_2^2}_{\approx \frac{1}{2\gamma b}} + 2w_0 w_1 \underbrace{\langle K_{1,b}, K_{1,\gamma b}\rangle}_{\approx \frac{1}{b(\gamma+1)}} \Bigg] \\
&= \frac{f(1)}{mb}\left( \frac{w_0^2}{2} + \frac{w_1^2}{2\gamma} + \frac{2w_0 w_1}{\gamma+1} \right)\big(1 + o(1)\big). && (34)
\end{aligned}
$$

For notational convenience, we define the *variance constant* $\Omega(\gamma)$ which depends solely on the step size $\gamma$:

$$
\Omega(\gamma) \equiv \frac{w_0^2}{2} + \frac{w_1^2}{2\gamma} + \frac{2w_0 w_1}{\gamma+1}. \tag{35}
$$

The MSE is the sum of the squared bias (see Equation (26)) and the variance:

$$
\begin{aligned}
\mathrm{MSE} &= \Big(\mathrm{Bias}[\widehat{f}_{\mathrm{jk}}(1)]\Big)^2 + \mathrm{Var}\big(\widehat{f}_{\mathrm{jk}}(1)\big) \\
&= \Big(-\gamma c_2 b^2\Big)^2 + \frac{f(1)}{mb}\Omega(\gamma)\big(1 + o(1)\big) \\
&= \gamma^2 c_2^2 b^4 + \frac{\Omega(\gamma) f(1)}{mb} + o(b^4) + o((mb)^{-1}). \tag{36}
\end{aligned}
$$

### C.4. Optimal Bandwidth Selection

To determine the optimal bandwidth $b_{\mathrm{opt}}$, we minimize the asymptotic Mean Squared Error (AMSE) derived in Equation (36). Let the AMSE be denoted by $J(b)$:

$$
J(b) = \gamma^2 c_2^2 b^4 + \frac{\Omega(\gamma) f(1)}{m} b^{-1}. \tag{37}
$$

Differentiating $J(b)$ with respect to $b$ and setting the derivative to zero yields the first-order condition:

$$
\frac{dJ}{db} = 4c_2^2 \gamma^2 b^3 - \frac{\Omega(\gamma) f(1)}{mb^2} = 0.
$$

Rearranging the terms to solve for $b$:

$$
4c_2^2 \gamma^2 b^5 = \frac{\Omega(\gamma) f(1)}{m} \implies b^5 = \frac{\Omega(\gamma) f(1)}{4m\gamma^2 c_2^2}.
$$

Thus, the optimal bandwidth choice that minimizes the MSE is:

$$
b_{\mathrm{opt}} = \left( \frac{\Omega(\gamma) f(1)}{4m\gamma^2 c_2^2} \right)^{1/5}. \tag{38}
$$

The optimal bandwidth $b_{\text{opt}}$ derived in Equation (38) depends on the unknown second-order bias constant $c_2 = f'(1) + f''(1)$ and the boundary density $f(1)$. In practice, we adopt the Beta$(c, 1)$ distribution as a flexible reference family to approximate the local behavior of the density near $p = 1$. The reference density is given by:

$$g(t; c) = ct^{c-1}, \quad t \in [0, 1], \quad c > 0. \tag{39}$$

The parameter $c$ characterizes the slope and curvature at the boundary. Given the sample $\{p_j\}_{j=1}^m$, the Maximum Likelihood Estimator (MLE) for $c$ is:

$$\hat{c} = -\frac{m}{\sum_{j=1}^m \ln(p_j)}. \tag{40}$$

Under this reference model, the boundary values of the density and its derivatives are:

$$g(1) = c, \qquad g'(1) = c(c - 1), \qquad g''(1) = c(c - 1)(c - 2).$$

Substituting these expressions into the definition of the bias constant $c_2$, we obtain:

$$\begin{aligned}
c_2 &\approx g'(1) + g''(1) \\
&= c(c - 1) + c(c - 1)(c - 2) \\
&= c(c - 1)\big[1 + (c - 2)\big] \\
&= c(c - 1)^2. 
\end{aligned} \tag{41}$$

Consequently, we estimate $c_2$ by plugging in the MLE $\hat{c}$:

$$\hat{c}_2 = \hat{c}^2(\hat{c} - 1). \tag{42}$$

Substituting $\hat{c}_2$ and $\hat{f}(1) \approx \hat{c}$ into Equation (38) provides the fully empirical optimal bandwidth used in our algorithm.

### C.5. Hyperparameter $\gamma$ Selection

While the theoretical derivation establishes the optimal order $O(m^{-1/5})$ for the bandwidth $b$, the practical performance of the JKBB estimator relies on the specific choice of the scaling factor $\gamma$ and the estimation of the bias constants. To address this, we detail a stability-based selection method followed by the complete algorithmic procedure.

To select the optimal $\gamma$ from a candidate grid $\Gamma$, we employ a stability scoring mechanism that balances statistical power with robustness. This approach involves performing subsampling (or bootstrapping) for each candidate $\gamma \in \Gamma$ to generate a distribution of estimates $\{\hat{\pi}_0^{(k)}(\gamma)\}$. From these subsamples, we compute the mean estimate $\mu_{\hat{\pi}_0}(\gamma)$ and a normalized standard deviation $\sigma_{\hat{\pi}_0}(\gamma)$, representing the estimator's instability. The optimal $\gamma^*$ is then determined by minimizing a penalized objective function:

$$\gamma^* = \arg\min_{\gamma \in \Gamma} \left( \mu_{\hat{\pi}_0}(\gamma) + \lambda \cdot \sigma_{\hat{\pi}_0}(\gamma) \right), \tag{43}$$

where $\lambda$ is a penalty parameter. This criterion simultaneously penalizes inflated estimates of $\pi_0$ (preserving power) and high variance (ensuring stability against data perturbations). In this paper, we set the $\lambda = 1$ by default.

### C.6. Consistency of the JKBB Estimator

We summarize the above analysis in the following theorem.

**Proposition C.1** (Consistency of the JKBB Estimator). *Suppose the density $f$ is twice continuously differentiable with bounded derivatives in a neighborhood of $p = 1$. Let the bandwidth sequence $b_m$ satisfy $b_m \to 0$ and $mb_m \to \infty$ as $m \to \infty$.*

*Then, the Jackknife-corrected Beta boundary estimator is consistent:*

$$\hat{f}_{\text{jk}}(1) \xrightarrow{P} f(1).$$

*Proof.* Based on the bias and variance derivations in Appendix C.3, the mean squared error of the estimator admits the expansion

$$\mathbb{E}\big[(\widehat{f}_{\text{jk}}(1) - f(1))^2\big] = O(b_m^4) + O\Big(\frac{1}{mb_m}\Big).$$

Under the assumed conditions $b_m \to 0$ and $mb_m \to \infty$, both terms vanish as $m \to \infty$. The convergence of the mean squared error to zero implies convergence in probability by Chebyshev's inequality, establishing consistency. $\qquad\square$

For reference, we next derive the convergence rate of the JKBB estimator. Balancing the squared bias term $O(b_m^4)$ and the variance term $O((mb_m)^{-1})$ in the mean squared error yields the optimal bandwidth scaling $b_m \asymp m^{-1/5}$. Substituting this choice into the MSE expansion gives

$$\mathbb{E}\big[(\widehat{f}_{\text{jk}}(1) - f(1))^2\big] = O\Big(m^{-4/5}\Big).$$

By Markov's inequality, this $L_2$ bound implies the following probabilistic convergence rate:

$$\Big|\widehat{f}_{\text{jk}}(1) - f(1)\Big| = O_p\Big(m^{-2/5}\Big).$$

## D. Detail Derivation for Adjusted Moment Estimator

We first define the raw moment estimator, $\hat{\pi}_{0,\text{raw}}$, as follows:

$$\hat{\pi}_{0,\text{raw}} = \frac{\hat{\mu}_1 - \hat{\mu}_{\text{test}}}{\hat{\mu}_1 - \hat{\mu}_0}, \tag{44}$$

where $\hat{\mu}_0, \hat{\mu}_1$, and $\hat{\mu}_{\text{test}}$ are the sample means from their respective datasets. By the Law of Large Numbers and the Continuous Mapping Theorem, $\hat{\pi}_{0,\text{raw}}$ is a consistent estimator of the true proportion $\pi_0$, i.e., $\hat{\pi}_{0,\text{raw}} \xrightarrow{p} \pi_0$.

Although $\hat{\pi}_{0,\text{raw}}$ is consistent for $\pi_0$, the procedure uses the reciprocal $1/\hat{\pi}_{0,\text{raw}}$ through the effective BH level $\alpha/\hat{\pi}_{0,\text{raw}}$. Since $x \mapsto 1/x$ is convex on $(0, \infty)$, Jensen's inequality yields

$$\mathbb{E}\left[\frac{1}{\hat{\pi}_{0,\text{raw}}}\right] \geq \frac{1}{\mathbb{E}[\hat{\pi}_{0,\text{raw}}]} \approx \frac{1}{\pi_0},$$

so the effective level is inflated on average: $\mathbb{E}[\alpha/\hat{\pi}_{0,\text{raw}}] \gtrsim \alpha/\pi_0$. This inflation is in the liberal direction and can lead to invalid FIR control in finite samples.

To quantify this bias, we perform a second-order Taylor expansion of the function $f(\hat{\pi}_{0,\text{raw}}) = 1/\hat{\pi}_{0,\text{raw}}$ around the true value $\pi_0$:

$$\frac{1}{\hat{\pi}_{0,\text{raw}}} \approx \frac{1}{\pi_0} - \frac{1}{\pi_0^2}(\hat{\pi}_{0,\text{raw}} - \pi_0) + \frac{1}{\pi_0^3}(\hat{\pi}_{0,\text{raw}} - \pi_0)^2.$$

Taking the expectation of both sides, we get:

$$E\left[\frac{1}{\hat{\pi}_{0,\text{raw}}}\right] \approx E\left[\frac{1}{\pi_0}\right] - \frac{1}{\pi_0^2}E[\hat{\pi}_{0,\text{raw}} - \pi_0] + \frac{1}{\pi_0^3}E[(\hat{\pi}_{0,\text{raw}} - \pi_0)^2].$$

Since $\hat{\pi}_{0,\text{raw}}$ is asymptotically unbiased, $E[\hat{\pi}_{0,\text{raw}} - \pi_0] \approx 0$. By the definition of variance, $E[(\hat{\pi}_{0,\text{raw}} - \pi_0)^2] \approx \text{Var}(\hat{\pi}_{0,\text{raw}})$. Thus, the equation simplifies to:

$$E\left[\frac{1}{\hat{\pi}_{0,\text{raw}}}\right] \approx \frac{1}{\pi_0} + \frac{\text{Var}(\hat{\pi}_{0,\text{raw}})}{\pi_0^3}.$$

The term $\text{Var}(\hat{\pi}_{0,\text{raw}})/\pi_0^3$ is the leading source of bias. It is a positive value, indicating that the naive estimator usually overestimates $1/\pi_0$. We define $g(\mu_0, \mu_1, \mu_{\text{test}}) = \frac{\mu_1 - \mu_{\text{test}}}{\mu_1 - \mu_0}$. According to the first-order Delta method, the variance of $\hat{\pi}_{0,\text{raw}}$ can be approximated as:

$$\text{Var}(\hat{\pi}_{raw}) \approx \left(\frac{\partial g}{\partial \mu_0}\right)^2 \text{Var}(\hat{\mu}_0) + \left(\frac{\partial g}{\partial \mu_1}\right)^2 \text{Var}(\hat{\mu}_1) + \left(\frac{\partial g}{\partial \mu_{\text{test}}}\right)^2 \text{Var}$$

$$= \frac{1}{(\mu_1 - \mu_0)^2}\left[\pi_0^2 \frac{\sigma_0^2}{n_0} + (1 - \pi_0)^2 \frac{\sigma_1^2}{n_1} + \frac{\sigma_{\text{test}}^2}{m}\right],$$

where $\sigma_i^2$ is the population variance and $n_i$ is the sample size.

Finally, to obtain the estimator for the variance, $\widehat{\mathrm{Var}}(\hat{\pi}_{0,\mathrm{raw}})$, we replace all unknown population parameters with their corresponding sample estimates:

$$\widehat{\mathrm{Var}}(\hat{\pi}_{0,\mathrm{raw}}) = \frac{1}{(\hat{\mu}_1 - \hat{\mu}_0)^2} \left[ \hat{\pi}_{0,\mathrm{raw}}^2 \frac{\hat{\sigma}_0^2}{n_0} + (1 - \hat{\pi}_{0,\mathrm{raw}})^2 \frac{\hat{\sigma}_1^2}{n_1} + \frac{\hat{\sigma}_{\mathrm{test}}^2}{m} \right]$$

Thus, we construct a corrected estimator, $\hat{\theta}_{1/\pi_0}$, which is designed to subtract the estimated leading bias term:

$$\hat{\theta}_{1/\pi_0} = \frac{1}{\hat{\pi}_{0,\mathrm{raw}}} - \frac{\widehat{\mathrm{Var}}(\hat{\pi}_{0,\mathrm{raw}})}{\hat{\pi}_{0,\mathrm{raw}}^3} \tag{45}$$

As established in Proposition 5.1, our estimator is consistent, meaning $\hat{\pi}_{\mathrm{mom}} \xrightarrow{p} \pi_{\mathrm{test}}$ as the sample sizes grow. This implies that $\hat{\pi}_0 \xrightarrow{p} \pi_0$, and by the continuous mapping theorem, the ratio $\frac{\pi_0}{\hat{\pi}_0}$ converges in probability to 1. This ensures that the FIR is controlled asymptotically.

### D.1. The Proof of Theorem 5.2

*Proof.* The PTDI procedure applies the BH procedure at level $\alpha$ to the scaled p-values $\tilde{p}_j = \hat{\pi}_{0,m} p_j$, where $\hat{\pi}_{0,m} = 1 - \hat{\pi}_{\mathrm{mom}}$. By Proposition 5.1, we have

$$\hat{\pi}_{0,m} \xrightarrow{P} \pi_{0,\mathrm{test}} = 1 - \pi_{\mathrm{test}}.$$

Therefore, the conditions of Proposition B.3 are satisfied with $\pi_{0,\infty} = \pi_{0,\mathrm{test}}$. Applying Proposition B.3 yields

$$\limsup_{n,m\to\infty} \mathrm{FIR}_{n,m} \leqslant \frac{\pi_{0,\mathrm{test}}}{\pi_{0,\infty}} \alpha = \alpha.$$

This completes the proof. $\square$

## E. Experimental Details

Recall from Equation (4) that the false identification rate (FIR) is defined as the expectation of the false identification proportion,

$$\mathrm{FIP} := \frac{\sum_{j=1}^m \mathbb{1}\{M_{n+j} = 0, j \in \mathcal{S}\}}{\max(|\mathcal{S}|, 1)}.$$

In our experiments, we report the empirical FIR by averaging FIP over 1000 times of Algorithm 1.

As for the experiment on VLMs, the generated token length is set to be 32.

### E.1. Data Split Setup

Unless otherwise specified (e.g., when varying $\pi_{\mathrm{test}}$), in each trial we randomly split the dataset into two equal halves. All non-members from one half are used to construct the calibration set $\mathcal{D}_{\mathrm{cal}}$, while the other half serves as the test set $\mathcal{D}_{\mathrm{test}}$. The detailed information of the constructed dataset is presented in Table 3.

For the experiment in Figure 4, in each trial, we further subsample the test set according to $\pi_{\mathrm{test}}$. Specifically, we select $\pi_{\mathrm{test}} \cdot |\mathcal{D}_{\mathrm{test}}^0|$ points from $\mathcal{D}_{\mathrm{test}}^1$ and $(1 - \pi_{\mathrm{test}}) \cdot |\mathcal{D}_{\mathrm{test}}^0|$ points from $\mathcal{D}_{\mathrm{test}}^0$, where $\mathcal{D}_{\mathrm{test}}^0$ and $\mathcal{D}_{\mathrm{test}}^1$ denote the non-member and member subsets of $\mathcal{D}_{\mathrm{test}}$, respectively.

For the adjusted moments estimator experiment in Figure 6, we again split the dataset into two equal halves, assigning one to $\mathcal{D}_{\mathrm{test}}$ and the other to $\mathcal{D}_{\mathrm{cal}}$.

### E.2. Adaptive FIR Controlling Procedures Baselines

The standard Benjamini-Hochberg (BH) method typically assumes a conservative null proportion of $\pi_0 = 1$. To improve statistical power while maintaining FIR control, adaptive procedures estimate the true proportion of null hypotheses, $\pi_0 = m_0/m$, directly from the data. We compare our approach against three established adaptive baselines:

| Dataset | Type | Member | Non-member | Total |
|---|---|---|---|---|
| WikiMIA | Test set | 460 | 398 | 858 |
| | Calibration set | / | 398 | 398 |
| | Original Dataset | 861 | 789 | 1,650 |
| ArXivTection | Test set | 381 | 393 | 774 |
| | Calibration set | / | 393 | 393 |
| | Original dataset | 762 | 786 | 1,548 |
| VL-MIA/Flickr | Test set | 150 | 150 | 300 |
| | Calibration set | / | 150 | 150 |
| | Original dataset | 300 | 300 | 600 |
| VL-MIA/DALL-E | Test set | 148 | 148 | 296 |
| | Calibration set | / | 148 | 148 |
| | Original dataset | 296 | 296 | 592 |
| XSum | Test set | 2,790 | 2,875 | 5,665 |
| | Calibration set | / | 2,875 | 2,875 |
| | Original dataset | 5,581 | 5,751 | 11,332 |
| BBC Real Time | Test set | 1,638 | 1,674 | 3,312 |
| | Calibration set | / | 1,674 | 1,674 |
| | Original dataset | 3,277 | 3,349 | 6,626 |

**BH-Storey.** Storey (2002) proposed estimating $\pi_0$ using a fixed tuning parameter $\lambda \in [0, 1)$. This method relies on the intuition that p-values corresponding to true null hypotheses are uniformly distributed, so those exceeding $\lambda$ are mostly nulls. The estimator is defined as:

$$\hat{\pi}_0^{\text{Storey}}(\lambda) = \frac{1 + \sum_{i=1}^m \mathbb{1}\{p_i \geq \lambda\}}{m(1 - \lambda)}, \quad \lambda \in (0, 1) \tag{46}$$

where $\mathbb{1}\{\cdot\}$ is the indicator function. A common choice is $\lambda = 0.5$. The estimated $\hat{\pi}_0$ is then plugged into the BH procedure to adjust the rejection threshold.

**Quantile-BH.** Benjamini & Hochberg (2000) introduced a graphical approach to estimate $\pi_0$ based on the quantile plot of sorted p-values. This method, often referred to as the "Lowest Slope Estimator", identifies a change point in the slope of the sorted p-values $p_{(1)} \leq \cdots \leq p_{(m)}$. The estimator is given by:

$$\hat{\pi}_0^{\text{Quant}}(k_0) = \frac{m - k_0 + 1}{m(1 - p_{(k_0)})}, \quad k_0 \in \{1, \ldots, m\} \tag{47}$$

Here, $S_k = \frac{1 - p_{(k)}}{m - k + 1}$ and the index $k_0$ is selected via a step-down stopping rule that searches for the first $k$ such that $S_k < S_{k-1}$, indicating a deviation from the linear behavior expected under the null and thus separating the signal tail from the null p-value distribution.

**BKY (Two-Stage BH).** Benjamini, Krieger, and Yekutieli (2006) proposed a two-stage linear step-up procedure (BKY) that avoids the need for manual tuning parameters like $\lambda$. In the first stage, the standard BH procedure is run at a stricter level $\alpha' = \alpha/(1 + \alpha)$. The number of rejections $r_1$ from this stage is used to estimate the null proportion:

$$\hat{\pi}_0^{\text{BKY}} = (m - r_1)/m \tag{48}$$

This estimate is then used in a second stage to scale the p-values.

*Table 4.* Bias and MSE of the JKBB estimator, evaluated with GPT-NeoX-20B on the ArxivTection dataset.

| Method | $\pi = 0.1$ | | $\pi = 0.3$ | | $\pi = 0.5$ | | $\pi = 0.7$ | | $\pi = 0.9$ | |
|---|---|---|---|---|---|---|---|---|---|---|
| | \|Bias\| | MSE | \|Bias\| | MSE | \|Bias\| | MSE | \|Bias\| | MSE | \|Bias\| | MSE |
| Perplexity | 0.048 | 0.018 | 0.006 | 0.011 | 0.031 | 0.007 | 0.054 | 0.007 | 0.068 | 0.007 |
| Zlib | 0.030 | 0.018 | 0.046 | 0.022 | 0.113 | 0.032 | 0.173 | 0.047 | 0.214 | 0.058 |
| MIN-K% | 0.055 | 0.019 | 0.011 | 0.010 | 0.006 | 0.006 | 0.009 | 0.003 | 0.015 | 0.001 |
| M-Entropy | 0.053 | 0.019 | 0.001 | 0.011 | 0.032 | 0.007 | 0.050 | 0.006 | 0.067 | 0.006 |

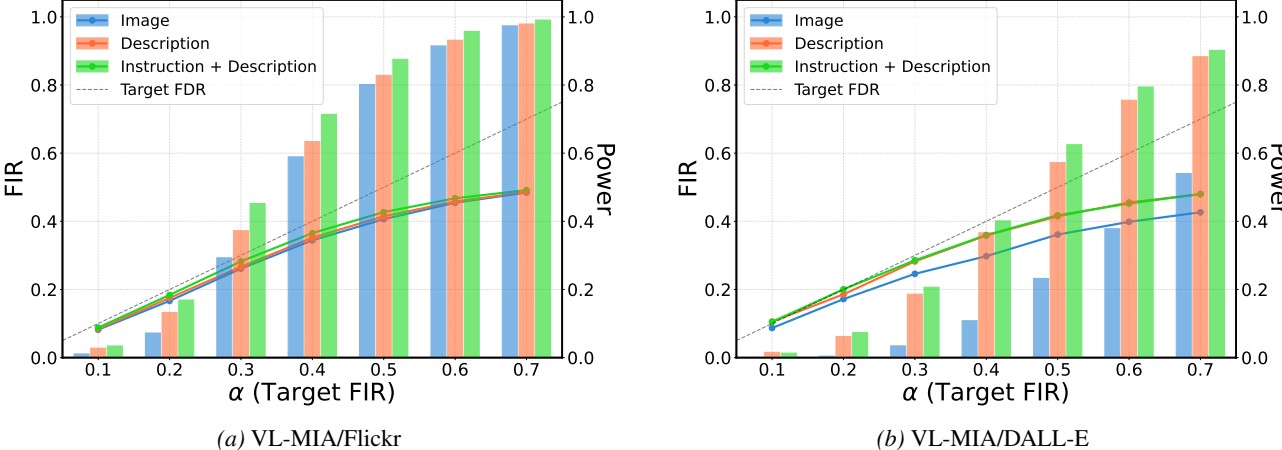

*(a)* VL-MIA/Flickr        *(b)* VL-MIA/DALL-E

*Figure 7.* FIR (solid lines) and power (bars) achieved by our method on LLaVA-1.5. Results are based on the MaxRényi-K% score computed from three types of inputs: image embeddings, generated descriptions, and instructions concatenated with descriptions.

## F. Additional Results

### F.1. Evaluation of JKBB Estimator

We evaluate the accuracy of the JKBB estimator for the test-set usage proportion $\pi_{\text{test}}$ on GPT-NeoX-20B using the ArXivTection dataset. For each target proportion $\pi \in \{0.1, 0.3, 0.5, 0.7, 0.9\}$, we construct test sets with the corresponding mixture of members and non-members, compute conformal p-values as in Equation (6), and apply the JKBB estimator defined in Equation (15). The reported results are averaged over multiple random trials.

Table 4 reports the absolute bias $|\mathbb{E}[\hat{\pi}_{\text{bdy}}] - \pi|$ and the mean squared error (MSE) $\mathbb{E}[(\hat{\pi}_{\text{bdy}} - \pi)^2]$ across different detection scores. Overall, the JKBB estimator exhibits small bias and low MSE over a wide range of $\pi$ values, demonstrating its robustness to varying test-set composition. n particular, detection scores that more effectively separate members from non-members (e.g., MIN-K%) result in more accurate estimation.

### F.2. Result on Vision Language Models

We further evaluate our method on training data identification for vision–language models (VLMs). Following prior work (Li et al., 2024c), we conduct experiments on the VL-MIA/Flickr and VL-MIA/DALL-E using LLaVA-1.5 (Liu et al., 2023). The detection score $T(X)$ is computed via the MaxRényi-K% statistic (Li et al., 2024c), with hyperparameters $K = 100$ and $\gamma = 0.5$. As shown in Figure 7, our method consistently controls the false identification rate (FIR), with the realized FIR remaining below the nominal level $\alpha$ across all settings.

# G. From Instance-wise Inference to Multiple Testing: The MIA Dilemma

## G.1. The Ideal: A Rigorous Single-Point Test

Given a data point $X$ and a trained target model $\theta_1$, membership inference attacks (MIAs) (Shokri et al., 2017; Yeom et al., 2018; Salem et al., 2019) aim to identify whether $X$ is one of the members in the training set $\mathcal{D}_{\text{train}}$. This type of privacy attack is often modeled as a statistical hypothesis testing problem (Ye et al., 2022; Carlini et al., 2022; Bertran et al., 2024; Zarifzadeh et al., 2024):

$$H_0 : X \notin \mathcal{D}_{\text{train}} \quad \text{v.s.} \quad H_1 : X \in \mathcal{D}_{\text{train}}. \tag{49}$$

Here, the null hypothesis ($H_0$) posits that $X$ is a non-member, meaning it was drawn from the same underlying data distribution as $\mathcal{D}_{\text{train}}$ but was not included in it. Conversely, the alternative hypothesis ($H_1$) posits that $X$ is a member of the training set.

To reject the null hypothesis, we compute membership inference attack (MIA) scores, such as the model's loss or confidence on data point $X$. For instance, let $T(X, \theta)$ represent the loss of $X$ produced by model $\theta$, then we can reject $H_0$ when $T(X, \theta) \leqslant \tau$ [1]. To control the type I error, which is identical to the false positive rate (FPR) at the sample level, we choose $\tau$ such that

$$\Pr_{\theta \sim \Theta_0} [T(X, \theta) \leqslant \tau] \leqslant \alpha \tag{50}$$

where $\Theta_0$ is the distribution over model parameters when trained on datasets that do not contain $X$ (under $H_0$), $\tau$ is the threshold used to define the rejection region. In practice, sampling $\theta$ typically requires training multiple reference models (Ye et al., 2022) or constructing Bayesian neural networks from a single reference model (Liu et al., 2025b).

However, estimating this reject region is challenging for large-scale models, such as ChatGPT or DALL-E, due to limited access to the training data distribution and the training algorithm, compounded by the prohibitively high computational cost of training (Zhang et al., 2025b).

## G.2. The Reality: A Heuristic with a Conceptual Flaw

Several studies on MIAs applied to large language models and vision-language models (Fu et al., 2024; Carlini et al., 2021; Shi et al., 2024; Zhang et al., 2025a) report true positive rates (TPR) at low FPRs using only MIA scores from the target model. This is typically achieved via a heuristic method: a single score threshold $\tau$ is determined on a calibration set of non-members to control the average FPR(Ye et al., 2022). Formally, this metric can be written as the expected Type I error:

$$\text{FPR} = \mathbb{E}_X \left[ \Pr[T(X, \theta_1) \leqslant \tau \mid H_0] \right]. \tag{51}$$

Herein lies a subtle but critical conceptual flaw. The average FPR conflates the overall error rate with the per-hypothesis Type I error rate, $\Pr[T(X, \theta_1) \leqslant \tau \mid H_0]$. As an average metric, it does not provide a probabilistic guarantee for any single inference. A low average FPR can mask a much higher error rate for specific subgroups of data, offering no reliable evidence for any individual decision.

Furthermore, in practical scenarios like copyright infringement litigation, where the goal is to identify a reliable set of members from many candidates, the average FPR remains inappropriate. This scenario is a classic multiple testing problem. In this setting, the objective is not to manage an average error rate over all non-members, but to control the fraction of incorrect claims among the discoveries made. This is precisely the quantity measured by the false identification rate (FIR), the statistically sound tool for this task.

The distinction between these metrics is critical in practice. Consider an audit of one million candidates containing only 1,000 true members. A method with seemingly excellent performance, such as a 0.1% average FPR and 80% TPR, would nevertheless yield approximately 999 false positives alongside 800 true positives. The resulting set's FIR would be an untenable 55.5%, meaning over half the presented evidence is incorrect. This starkly demonstrates that average FPR is a fundamentally inadequate and misleading metric for ensuring the credibility of membership inference claims in a legal or auditing context.

---

[1]Lower loss suggests that $X$ was likely to be a part of the training set.

