# OpenReview forum: "Provable Training Data Identification for Large Language Models"
_ICML.cc/2026/Conference — ICML 2026 regular_

### Official Review · Reviewer_mWwp · 2026-03-05

**Soundness:** 2
**Presentation:** 3
**Significance:** 3
**Originality:** 3
**Overall Recommendation:** 4
**Confidence:** 3

**Summary:**

This work introduces the notion of training data identification as a set-level inference problem rather than a traditional instance-wise classification problem. The main contribution is the Provable Training Data Identification (PTDI) method, which is a distribution-free approach to the proposed problem that attempts to maximize the power of the test under a given False Identification Rate (FIR) budget $\alpha$.

The proposed method requires only a confirmed set of non-members. Under i.i.d. assumptions, the authors prove that their algorithm asymptotically respects the FIR budget $\alpha$. Experimental evidence supports the claim that the proposed method provides reliable FIR control while also achieving higher power compared to baselines.

**Compliance With Llm Reviewing Policy:**

Affirmed.

**Final Justification:**

The rebuttal addressed my concerns about whether the experimental design is appropriate.
I think a more careful framing of what is "provable" will improve the flow and the authors agreed to do so.
For these reasons, I am leaning positive.
However, due to my relative lack of familiarity with the area, I cannot confidently give a stronger recommendation.

**Key Questions For Authors:**

1. Are there scenarios where it would be reasonable to expect calibration data that is distributionally similar to the test set?
1. Are there other (perhaps trivial) baselines that one could consider?

**Limitations:**

yes

**Strengths And Weaknesses:**

## Soundness

The theoretical results are supported by rigorous proofs, although I have not checked the proofs in significant detail.

The experiments seem somewhat reasonable, with ample variations of datasets/architectures that confirm the proposed method respects the FIR budget $\alpha$. However, comparison with baselines seem a bit lacking, with only a single baseline studied (KTD). Also, the requirement of a calibration set of unseen but distributionally similar data to the test set may be somewhat strong of a requirement.

## Presentation

I found the paper to be decently well-organized and easy to follow.

## Significance

Training data detection is an increasingly important problem, and progress in this direction has implications for data poisoning / copyright infringement / etc. The experiments confirm that the proposed method more closely respects the given FIR budget compared to the baseline while achieving higher power. As mentioned, comparison to other baseline techniques is a bit lacking.

The title is slightly (perhaps unintentionally) misleading, as the main theoretical result is about respecting the FIR budget under (very) ideal i.i.d. and asymptotic settings in which the number of data points tends to infinity. No guarantees are given in terms of the power, so the algorithm is not fully “provable”.

## Originality

The formulation of training data identification as a set-level question seems novel and quite reasonable to me. Although not groundbreaking, the proposed method seems principled and avoids the need to train multiple reference models like certain Membership Inference Attacks (MIA), which is very desirable given the computational cost of training SOTA models.

---

> ### Author Rebuttal · Authors · 2026-03-31
>
> Thank you for your positive assessment. We provide detailed responses below.
>
> ### **1. Other baselines beyond KTD [W1, Q2]**
>
> We sincerely thank the reviewer for this question. We would like to clarify our experimental design and explain why our baseline selection is both appropriate and comprehensive for this specific problem setting:
>
> Very few existing works attempt to provide theoretical guarantees for training data detection. KTD (Hu et al., 2025) is the closest work that also targets FIR-controlled identification, making it the most relevant direct comparison. This scarcity also highlights the importance of our work in establishing principled methods for this problem.
>
> Existing heuristic MIA methods (e.g., Perplexity, MIN-K%, M-Entropy) are **complementary** to our framework rather than competing baselines. Our method is score-agnostic and integrates with these scores to provide FIR guarantees on top of them. Our experiments already incorporate these scores as inputs to our pipeline (Figures 1).
>
>  Beyond KTD, Table 1 compares our method against **four established adaptive BH procedures** within our framework. Our method outperforms all of them consistently across all model-score combinations.
>
> ### **2. Scenarios for distributionally similar calibration data [W2, Q1]**
>
> The requirement for an unseen, distributionally similar calibration set is indeed an important assumption, but it is feasible in the primary real-world applications our method targets:
> - **Copyright auditing**: Unreleased works from the same corpus (e.g., unpublished articles by the same author, unreleased films from the same studio).
> - **Benchmark contamination**: New evaluation items from the same benchmark distribution (e.g., newly created math problems).
> - **Private data**: Internal documents that have never been connected to the internet.
>
> ### **3. Scope of provable [W3]**
>
> We agree "provable" primarily refers to FIR control — that is, we guarantee the selected subset is statistically reliable with at most $\alpha$ fraction of false identifications. Regarding power, it depends not only on our BH procedure but also on the actual data distribution and the detection score's discriminative ability, making theoretical power guarantees unrealistic in general. Our FIR guarantee enables the identified set to serve as credible evidence in downstream applications, such as copyright litigation. We will clarify in the final version abstract that "provable" refers to FIR control.

---

> > ### Author Rebuttal · Reviewer_mWwp · 2026-03-31
> >
> > Thanks for the clarifications. I think including these discussions will improve the paper's presentation. I will discuss my score with the other reviewers/AC, given my better understanding of the scope of contributions.

---

> > > ### Author Response · Authors · 2026-04-04
> > >
> > > We are glad that our response helped clarify and address your concerns. We will incorporate these clarifications into the final version. Thank you again for the constructive feedback and careful review.

---

### Official Review · Reviewer_3PSv · 2026-03-12

**Soundness:** 2
**Presentation:** 3
**Significance:** 2
**Originality:** 3
**Overall Recommendation:** 4
**Confidence:** 2

**Summary:**

This paper studies training data identification for large language and vision-language models. Instead of treating the problem as instance-wise membership classification, the authors formulate it as a set-level inference task and propose Provable Training Data Identification (PTDI), which aims to identify a subset of samples while controlling the false identification rate (FIR). The method constructs conformal p-values from existing detection scores, estimates the proportion of training samples in the test set using a JKBB estimator, and applies a BH-style selection procedure. The authors provide theoretical analysis of FIR control and evaluate the framework across several LLM and VLM models using multiple detection scores.

**Compliance With Llm Reviewing Policy:**

Affirmed.

**Final Justification:**

The rebuttal has solved my concerns through experiments. I will raise the score accordingly

**Key Questions For Authors:**

See Weaknesses.

**Limitations:**

yes

**Strengths And Weaknesses:**

Strengths:
1. The paper targets a practically meaningful problem: identifying whether a model was trained on particular data, with theoretical guarantees.
2. The overall pipeline is coherent, and the paper provides theoretical discussion for the validity of the procedure rather than relying purely on empirical threshold tuning.
3. The experiments cover multiple model families, datasets, and both LLM/VLM settings, and include comparisons on error control and power.

Weaknesses:
1.  The proposed framework is designed to be score-agnostic and can be combined with existing detection scores. However, recent studies suggest that membership inference signals in LLMs are often weak and unstable [1,2]. When the underlying detection score provides limited separability between members and non-members, the proposed statistical calibration may yield only limited practical gains.
2. The main FIR-control theorem assumes the calibration and test covariates are i.i.d., which effectively requires the non-member calibration set to be distributionally matched to the test set. This assumption is rather strong in practical LLM auditing, where the training distribution is opaque and distribution shift is common. Moreover, Theorem 3.1 is asymptotic rather than a finite-sample guarantee.
3. The JKBB estimator further relies on sufficient smoothness of the p-value density near p=1 to justify its boundary bias correction. This technical condition is hard to verify in practice and may be fragile in finite-sample settings, especially when calibration data are limited.


[1] Do Membership Inference Attacks Work on Large Language Models? COLM 2024

[2] Nob-MIAs: Non-biased Membership Inference Attacks Assessment on Large Language Models with Ex-Post Dataset Construction

---

> ### Author Rebuttal · Authors · 2026-03-31
>
> We thank you for your insightful comments. Detailed responses are provided below.
>
>
> ### **1. Reliable identification under weak signals [W1]**
> Our method is not about finding as many members as possible, but about finding reliable members with statistical guarantees. Even when the underlying detection signal is weak, our method still selects a subset for which the FIR is controlled at level $\alpha$. In legal and audit applications, presenting 50 correctly identified samples with 95% credibility is far more valuable than 200 samples with an unknown error rate. Without our framework, no such guarantee can be provided.
>
> We further evaluate our method on the reviewer-suggested challenging MIMIR [1] benchmark, using Pythia-6.9B as the target model and Min-K%++ [2] as the detection score. The results below show that the realized false identification rate remains consistently below the target level $\alpha$ across all datasets.
>
> | Dataset   | $\alpha$=0.1 | $\alpha$=0.15 | $\alpha$=0.2 |
> | --------- | -----------: | ------------: | -----------: |
> | GitHub    |       0.0710 |        0.1078 |       0.1446 |
> | Pile-CC   |       0.0574 |        0.0812 |       0.1065 |
> | ArXiv     |       0.0653 |        0.0939 |       0.1212 |
> | Wikipedia |       0.0786 |        0.1117 |       0.1416 |
>
>
> ### **2. i.i.d. assumption and asymptotic guarantee [W2]**
>
> - **i.i.d. assumption**: The i.i.d. requirement is between the calibration non-members and the test non-members; it is not necessary for us to know the training data distribution. In practice, this is enforced by construction, as the auditor samples the calibration set from the same domain as the test data. We acknowledge that this assumption can still be strong, especially when the test set is highly heterogeneous. However, this requirement is still considerably milder than those in many prior MIA methods, which often assume access to the training data distribution and require a matched reference model. Our method only requires an unseen calibration set aligned with the test domain. This is also weaker than the requirements of several heuristic MIA methods, which typically rely on an additional validation set for hyperparameter tuning, such as Min-K% [3], Min-K%++ [2], and MaxRényi-K% [4]. We discuss this limitation explicitly in the Limitations section, and consider handling distribution shifts as a promising direction for future work.
>
> - **Asymptotic vs. finite-sample**: The asymptotic nature arises from our JKBB estimator, which is introduced to **improve power**. If we instead used a simpler estimator (e.g., Storey's) for a finite-sample guarantee, the resulting procedure would be overly conservative with substantially lower power, limiting its practical utility. Crucially, our empirical results provide strong finite-sample evidence: (1) Figures 1, 4 demonstrate FIR control holds tightly at practical hundreds of sample sizes  ; (2) Figure 5 shows FIR remains controlled even with a small calibration-to-test ratio.
>
>
> ### **3. JKBB smoothness assumption [W3]**
>
> Thank you for this insightful technical point. We agree that the smoothness assumption near p=1 is important, and we address it from theoretical, empirical, and finite-sample perspectives:
> - **Theoretical Justification.** The marginal p-value density is a mixture of non-members and members $f_{test}(p) = (1-\pi_{test})f(p|M=0) + \pi_{test}f(p|M=1)$. The non-member density is a uniform distribution.  For members, the density $f(p|M=1)$ is derived from detection scores (e.g., perplexity, entropy), which are continuous random variables produced by continuous logit spaces. Consequently, the member p-value density also decays smoothly near p=1, theoretically satisfying the twice-differentiable condition.
> - **Empirical Verification.** We further provide empirical verification via **p-value distribution histograms** in Figure 3 of [link](https://anonymous.4open.science/r/PTDI_r/). These plots clearly demonstrate that real-world LLM/VLM p-value densities decay smoothly near p=1, empirically validating our local smoothness assumption.
> - **Robustness with Limited Calibration Data.**  We explicitly evaluated the finite-sample fragility in our ablation study. Figure 5 shows that our method still controls the FIR below the target level even when the calibration set is only 10% the size of the test set.
>
> [1] Do Membership Inference Attacks Work on Large Language Models? (COLM 2024)
>
> [2] Min-K%++: Improved Baseline for Detecting Pre-Training Data from Large Language Models (ICLR 2025)
>
> [3] Detecting pretraining data from large language models. (ICLR, 2024)
>
> [4] Membership Inference Attacks against Large Vision-Language Models (ICLR, 2025)

---

> > ### Author Rebuttal · Reviewer_3PSv · 2026-04-03
> >
> > The rebuttal has solved my concerns through experiments. I will raise the score accordingly

---

> > > ### Author Response · Authors · 2026-04-04
> > >
> > > We are glad that our response helped resolve your concerns. We will incorporate these experimental results and clarifications into the final version. Thank you again for the helpful comments and thoughtful evaluation.

---

### Official Review · Reviewer_txNg · 2026-03-12

**Soundness:** 3
**Presentation:** 3
**Significance:** 4
**Originality:** 3
**Overall Recommendation:** 4
**Confidence:** 3

**Summary:**

This paper studies the problem of training data identification, which is closely related to membership inference attacks (MIA). The goal is to determine whether a given sample was used during model training.
The main idea is to use tools from conformal prediction. For a given input, the method computes a conformal p-value and derives a statistic related to the prediction set size (PSS). The authors argue that training samples and non-training samples tend to produce different distributions of these statistics.
Based on this observation, the paper proposes a testing procedure that identifies whether a sample is likely to be a training member. The paper analyzes the statistical behavior of these quantities and evaluates the method using metrics such as False Identification Rate (FIR) and Power, which correspond to controlling the error rate while maximizing the probability of correctly identifying members.
The authors also present theoretical analysis about the statistical properties of the proposed quantities and run experiments comparing their method with several existing membership inference baselines.
Overall, the paper tries to connect membership inference with a statistical hypothesis testing framework, and to provide theoretical insight into why certain statistics may help detect training data.

**Compliance With Llm Reviewing Policy:**

Affirmed.

**Final Justification:**

I appreciate the rebuttal for improving clarity on the guarantee, FIR, and additional experimental details. However, it does not provide strong new evidence to justify a higher rating. The “provable” guarantee does not directly reflect detection accuracy, and the advantage over standard metrics like FPR is still not clearly demonstrated. The concern about temporal distribution shift is also not fully resolved. Therefore, I keep my original weak accept rating, and I do not have enough confidence to increase it further.

**Key Questions For Authors:**

Q1. The paper presents several theoretical results, but it is not entirely clear how these results translate into guarantees for training data identification accuracy. Could the authors clarify precisely what aspect of the method is provably guaranteed?

Q2. FIR and Power seem closely related to the standard metrics FPR and TPR. Is there any practical advantage of using this formulation beyond a change in statistical terminology?

Q3. Could you test some simple baselines such as a bag-of-words classifier or other dataset-only features to verify that the benchmark itself does not already allow easy separation between member and non-member samples?

Q4. Could the authors clarify the computational cost of the proposed method and whether it remains efficient for large models?

Q5. The proposed decision rule appears closely related to the thresholding strategy discussed in Ye et al. (2022). In both cases, a statistic is compared against a threshold determined by a target error rate. Could the authors clarify what the key methodological difference is between the proposed approach and prior threshold-based methods?

**Limitations:**

- The method implicitly assumes that member and non-member samples follow the same underlying distribution and that the model outputs are reasonably calibrated. In practice, both assumptions may be violated. For example, the paper suggests constructing non-member data using samples collected after the training cutoff date, which may introduce temporal distribution shift. In addition, prediction set size and related statistics depend on the model’s probability outputs, which may be poorly calibrated in large generative models. If calibration or data distribution changes, the observed differences between member and non-member samples may partly reflect these factors rather than model memorization.

- The paper presents several theoretical results about the statistical behavior of conformal scores and prediction set size. However, these results mainly describe distributional properties and bounds, rather than providing a direct guarantee on membership detection accuracy.

**Strengths And Weaknesses:**

Strengths

1. The paper connects membership inference with tools from conformal prediction and hypothesis testing. This provides a new way to think about the problem, and the theoretical discussion helps explain why certain statistics may behave differently for training and non-training samples.

2. The paper attempts to analyze the statistical properties of the proposed scores and discusses how these quantities relate to distinguishing member and non-member samples. Even though some assumptions may be strong, the effort to provide theoretical reasoning is valuable.

3. Training data identification is an important topic because it relates to privacy, data ownership, and model auditing. Understanding when models reveal information about their training data is an important research direction.

Weaknesses

1. The theoretical results mainly describe statistical properties of the proposed scores and provide bounds on distribution differences. However, they do not directly give a guarantee on membership detection accuracy. Because of this, the claim of “provable training data identification” seems somewhat stronger than the theoretical results provided in the paper.

2. The paper uses terms such as FIR and Power, but these appear to correspond closely to the standard metrics FPR and TPR used in membership inference. As a result, the proposed evaluation framework may largely be a change of terminology rather than a fundamentally different formulation of the problem.

3. One concern is the assumption that member and non-member samples follow the same underlying distribution. In practice, the paper suggests constructing non-member sets using data collected after the model’s training cutoff date (e.g., recent news articles or images). However, such data naturally introduces temporal distribution shift. For example, new events, vocabulary, or visual styles may appear over time. In this situation, it may already be possible to distinguish the two sets using simple dataset-level signals (e.g., bag-of-words features), without relying on model memorization. This raises the question of whether the observed performance truly reflects training data identification or partly captures distribution differences between the datasets.

4. The method relies on conformal p-values and prediction set size, which appear to require computing statistics over the model’s output distribution. For models with large output spaces (e.g., large language models with very large vocabularies), this may introduce additional computational overhead.

---

> ### Author Rebuttal · Authors · 2026-03-31
>
> Thank you for your positive assessment and thoughtful questions. We provide detailed responses below.
> ### **1. Scope of the provable guarantee [W1, Q1]**
>
> We clarify that "provable" refers specifically to the **False Identification Rate (FIR)** of the selected set (Theorem 3.1). We theoretically guarantee that if an auditor uses our method to output a list of flagged training data, the expected proportion of false positives within that specific list  is strictly bounded by the user-defined $\alpha$. This guarantee enables practical applications such as presenting credible evidence in copyright litigation. We will clarify the scope more precisely in the final version
> ### **2. FIR vs. FPR terminology [W2, Q2]**
> FIR and FPR are **fundamentally different** metrics, not a change of terminology. FPR is an average per-instance error rate across all non-members. FIR is the expected fraction of errors **within the selected set**. The distinction is critical in practice: consider a method with 0.1% FPR and 80% TPR applied to 1M candidates with 1K true members. It produces ~999 false positives alongside 800 true positives, yielding $\text{FIR} \approx 55.5\%$. Controlling FIR at 5% guarantees that at most 5% of the reported training set is false, a much stronger and more actionable guarantee for legal/audit settings. We detail this distinction mathematically in Appendix G.
> ### **3. Temporal distribution shift [W3, Q3]**
> Thank you for raising this insightful point. We would like to clarify that our theorem does **not** assume that member and non-member samples follow the same distribution. Rather, the temporal distribution shift issue concerns whether a detection metric, such as AUC, can faithfully reflect the model memorization. Our framework is **score-agnostic**: such shifts primarily affect the quality of the underlying detection scores, not the validity of our statistical procedure.
>
> While the Bag-of-Words baseline could test for basic vocabulary differences, we believe the most direct way to address the core concern is to evaluate on a benchmark that is entirely free of time cutoffs. Therefore, rather than analyzing dataset-level features on a temporally split dataset, we conducted supplementary experiments on MIMIR-Github, where member and non-member construction is completely independent of temporal splits.
>
> ### **4. Computational cost [W4, Q4]**
> Our method adds **negligible overhead** beyond computing detection scores. Given pre-computed scores for $n+m$ samples, the JKBB estimation is $O(m)$ and the BH procedure is $O(m \log m)$. The dominant cost remains a single forward pass per sample, as in existing training-data detection methods. Moreover, our conformal p-values require only rank comparisons on the precomputed scores, without recomputation over the full output vocabulary.
>
> ### **5. Difference from Ye et al. (2022) [Q5]**
>
> The key difference is twofold:
> 1. **Error Control Target**: Ye et al. control the point-wise Type I error, while our method controls the set-level FIR.
> 2. **Reference Models**: Ye et al. require training multiple reference models to estimate the null distribution for every individual data point. This is computationally infeasible for LLMs/VLMs. Our method is distribution-free and requires no reference models.

---

> > ### Author Rebuttal · Reviewer_txNg · 2026-04-02
> >
> > Thank you for the rebuttal. While it improves clarity, it does not sufficiently address my main concerns.
> >
> > The distinction between FIR and FPR still feels largely conceptual and not fully justified beyond conditioning on the selected set. The nature of the claimed guarantee also remains unclear. The response to temporal distribution shift does not directly address its impact on validity.
> >
> > Overall, I do not find enough new evidence or justification to change my evaluation. I will keep my original score.

---

> > > ### Author Response · Authors · 2026-04-04
> > >
> > > Thank you for taking the time to engage with our rebuttal. We hope the following clarifications resolve your remaining concerns:
> > >
> > > ### **1. The Practical Necessity of FIR over FPR**
> > >
> > > We agree that FIR is conditioned on the selected set, whereas FPR is evaluated against the full null pool. Specifically, FIR measures the expected fraction of false positives among the selected set, while FPR measures the fraction of null items that are incorrectly flagged. In practical settings such as copyright litigation, this distinction is crucial because the legal risk is determined by the reliability of the reported list, not by the average error rate over all non-members.
> > >
> > > In copyright litigation, the judicial system is solely concerned with the specific objects accused of infringement (the selected set). FIR directly answers the legal question: "**Given the specific list of targets being sued, what is the probability that we are making a false accusation?**" If we merely rely on FPR, even an excellent rate (e.g., 1%) applied to a massive internet-scale search (the full null pool) will generate thousands of false positives.  For example, if an auditor tests 1,000,000 candidates where only 1,000 are true members, a method with an excellent 1% FPR and 80% TPR will yield 9,990 false positives and 800 true positives. As a result, 92% of the items on the final evidence list presented to a judge would represent false accusations.
> > >
> > > FIR directly addresses this by **guaranteeing the purity of the final output list**. By controlling FIR at a chosen threshold (e.g., 5%), we guarantee that the specific list of flagged items meets the corresponding accuracy requirement (e.g., 95%). Therefore, our provable guarantee specifically means that the expected proportion of false identifications within the final reported list is strictly bounded by the user-defined threshold ($\alpha$),  which is necessary for legal or formal auditing contexts.
> > >
> > > ### **2. Regarding the Temporal Distribution Shift in Benchmarks**
> > >
> > > We agree that prior work has criticized some MIA benchmarks for introducing temporal distribution shift between member and non-member sets, which can inflate the empirical attack performance [1]. In the final version, we will provide practical guidance on carefully constructing a representative calibration set by evaluating corpus-level similarity statistics, such as n-gram overlap distributions, to avoid unintentional distributional shifts.
> > >
> > > At the same time, we clarify that this concern does not directly affect the validity of our method. The above criticism mainly concerns whether a benchmark may overestimate the empirical performance of a particular detection score. This concern is distinct from the validity of our set-level FIR guarantee, which depends on the assumptions stated in our theorem. To further empirically address the benchmark concern, we add new experiments on MIMIR [1], which is carefully constructed to reduce unintentional distribution shift between member and non-member sets. We use Pythia-6.9B as the target model and Min-K%++ [2] as the detection score. The results below provide additional empirical evidence that our method continues to control FIR on a benchmark specifically designed to mitigate this concern.
> > >
> > >
> > > | Dataset   | $\alpha=0.1$ | $\alpha=0.15$ | $\alpha=0.2$ |
> > > | --------- | -----------: | ------------: | -----------: |
> > > | GitHub    |       0.0710 |        0.1078 |       0.1446 |
> > > | Pile-CC   |       0.0574 |        0.0812 |       0.1065 |
> > > | ArXiv     |       0.0653 |        0.0939 |       0.1212 |
> > > | Wikipedia |       0.0786 |        0.1117 |       0.1416 |
> > >
> > > Thank you again for the helpful comments. We will incorporate these clarifications and experimental results into the final version.
> > >
> > >
> > > [1] Do Membership Inference Attacks Work on Large Language Models? (COLM 2024)
> > >
> > > [2] Min-K%++: Improved Baseline for Detecting Pre-Training Data from Large Language Models (ICLR 2025)

---

### Official Review · Reviewer_3Gd6 · 2026-03-13

**Soundness:** 2
**Presentation:** 3
**Significance:** 4
**Originality:** 3
**Overall Recommendation:** 3
**Confidence:** 3

**Summary:**

This theoretical paper integrates conformal prediction (called conformal inference) into the classical case of MIAs for LLMs and VLMs. This paper modifies the perspective of classical MIA, shifting the focus from instance-wiseclassification to set-level identification, with the aim of gaining real control over false positives.

For this purpose, the paper introduces a new algorithm called "Provable Training Data Identification" (PTDI), which, based on detection scores, computes scaled conformal p-values. Then, these p-values are used to obtain an adaptive rejection threshold below which the identification of data in the test set used in the training set has a type 1 error of $\alpha$.

The empirical results, that cover 8 LLMs/VLMs, 6 datasets and 5 detection scores, show an improvement in the false identification rate (FIR) and the power, newly defined in this paper. This complements the main theorem, which states that the upper bound of the FIR is bounded by $\alpha$ when the size of the test set tends to infinity.

**Compliance With Llm Reviewing Policy:**

Affirmed.

**Final Justification:**

Although I still have some reservations about the strength of the empirical evidence and the positioning relative to prior conformal prediction work, the rebuttal clarified the experimental inconsistencies, added information on confidence intervals and assumptions, and overall addressed my main concerns sufficiently to improve my evaluation to a more positive final recommendation of weak accept.

**Key Questions For Authors:**

Theorical part:
===

1. You use a kernel density estimation  (Eq. 11) but you use another one in your proof (Eq. 22) then reuse the original one (Eq. 27). Is there an equivalence?

2. Is the following sentence "for a broad class of alternatives, the density of member p-values is suppressed nearthe upper boundary p = 1" an assumption or a statement? It seems that given the difficulties of MIAs on current LLMs to well differentiate the data, it should proably be specified that this is a strong assumption. As a consequence, an empirical validation study is needed (see Empirical part below).

3. Which part before the theorem are your contributions, and which are intermediate definitions ? To which extent is it specially designed for LLMs/VLMs, or generic to any MIA ? Is it the fact of having used a mixture of models to approximate $\pi_\text{test}$ ?

Empirical part:
===

4. Figure 3 seems problematic for two reasons: (i) confidence intervals are missing, the improvement seems weak and with CIs maybe even non-significant, and (ii) we have a power which is close to 0.95 for $\alpha$ equal 0.10, 0.15 and 0.20. But when we look at Figure 1. (a), the powers are not above 0.3. Why does GPT-2 outperform so much ?
5. Table 1 lacks confidence interval or standard deviation. Is it significant?

6. On Figure 5, there is no confidence intervals, putting the standard deviation does not make sense because we expect that there is no symmetry given the theorem. Does the upper CI cross the target FIR?

7. Finally to complete Figure 4., a study of the assumption on the restricted number of positives around p=1 may be needed. Up to how many can we accept? At what distance? Etc.

**Limitations:**

The part about the assumption should be further analyzed.

**Strengths And Weaknesses:**

Strengths
===

- The theoretical framework is convincing. The proofs seems correct with reasonable assumptions. In particular, the main theorem appears to be well supported because it makes natural sense in the context of conformal prediction.

- The article is generally clear. The paper is well written, especially the mathematical sections, which are clear and easy to follow.

- The future directions seem significant. The paper is original in its perspective, as it lies at the intersection of MIA with LLMs/VLMs and conformal prediction, with potential impact in real-world scenarios.

Weaknesses
===

- The empirical evaluation is a weak point of this paper, due to the lack of confidence intervals for very close values (very fine scale on the vertical axis). This makes it difficult to compare existing techniques with those presented.

- The positioning in relation to the literature is still incomplete, as there is no related work section in the paper (it is in the appendix) and there is no paragraph on conformal prediction (which is a very rich field) and its applications in the considered context. It is difficult to assess the novelty in relation to what has already been done, as well as the positioning.

- The novelty mainly comes from the a combination of existing technqiues. In the theoretical part, it is difficult to see the mathematical novelty from what already exists (e.g., Eq. 4 (FIR), Eq. 5 (Power), Eq. 8, Eq. 13, etc.). So, despite the original point of view, the mathematical originality seems limited and the proofs and approximation techniques used seem fairly standard in conformal prediction.

---

> ### Author Rebuttal · Authors · 2026-03-31
>
> We thank the reviewer for the constructive review.
> ### **1. Confidence intervals for empirical results [W1, Q4, Q5, Q6]**
> Thank you for your constructive advice. All of our experimental results are averaged over 1000 experiments, ensuring the significance of our results. For reference, the corresponding confidence intervals for Figs 3 and 5 are available in Figs 1 and 2 of  [link](https://anonymous.4open.science/r/PTDI_r/). The standard error of Table 1 in our manuscript is provided in  Table 1. These results consistently show that our method controls FIR while achieving higher power.
> ###  2. Cross-setting power differences.
> We apologize for a caption error: Fig 3 reports results on BBC Real Time (as correctly stated in Sec 4.2), not WikiMIA. Beyond the dataset/model difference, the higher power in Fig 1 (b) relative to Fig 1(a) mainly stems from two factors:
> - **The detection score**: knockoff statistic is a white-box score, while the detection scores like Perplexity are black-box.
> - **The training setting**: Fig 1 uses pre-trained models, while Fig 2 and 3 follow KTD, where member test data are fine-tuned by the model.
> We will correct the caption in the final version
> ### **3. Related work positioning [W2]**
> Due to space limits, the full related work is deferred to the appendix, but we will add a concise discussion of conformal selection to the main text. Specifically, Jin et al. [1] is inapplicable because it requires exchangeable calibration and test sets. Bates et al. [2] studies Storey-type scaling for conformal p-values, whereas our JKBB estimator achieves higher power than Storey’s.
> ### **4. Clarification of Novelty [W3, Q3]**
> We clarify that the main **contribution** of this work is proposing an effective method with theoretical guarantees to address the critical problem of training data identification in large-scale models. In particular, our **novelties** are threefold:
> - **Problem formulation**: We formulate training data identification as a multiple-hypothesis testing problem, thereby enabling rigorous FIR control. Before this work, the MIA community argued that proving a model was trained on specific data was intractable [3]. **Our formulation provides the first feasible path forward.**
> - **Methodology**: We propose two novel estimators to achieve high power: (1) the **JKBB estimator**, which combines Beta boundary kernels with jackknife bias correction to mitigate the boundary overestimation of Storey's estimator; and (2) the **Adjusted moment estimator**, which leverages available member information to further improve power.
> - **Theoretical guarantee**: We establish comprehensive theoretical foundations for our proposed estimators. Specifically, we derive the bias correction and the task-specific optimal bandwidth, prove the consistency of both estimators, and explicitly establish formal FIR control guarantees.
> > **Regarding Q3 (generality & mixture models):** Our method applies generically to any detection score, but is uniquely practical for LLMs/VLMs where training reference models is prohibitive. We do not use a mixture of models; we simply model the p-values as a mixture distribution of members and non-members.
> ### **5. Notation inconsistency [Q1]**
> We thank the reviewer for their careful reading, and we sincerely apologize for the notational inconsistency. The two kernel definitions are strictly equivalent under parameter substitution, affecting no subsequent proofs.
>
> Specifically, $K_A(t; b) = (\frac{1}{b}+1)t^{1/b}$  matches $K_B(t; b) = \frac{1}{b}t^{\frac{1}{b}-1}$  via $b' = \frac{b}{b+1}$. As $b \to 0$, $b' = b - b^2 + O(b^3) \approx b$, so the difference only affects higher-order terms. Thus, the leading-order bias/variance, optimal bandwidth $O(m^{-1/5})$, and all guarantees remain unchanged. We will unify the notation in the final version.
>
> ### **6. On the member p-value distribution near p=1 [Q2, Q7]**
> Density suppression near $p=1$ is a **sufficient condition for improved power**, not a requirement for FIR control. Thm 3.1 guarantees FIR $\le \alpha$ as long as $f_{\text{test}}(1) \ge 1-\pi_{\text{test}}$ . This inequality follows directly from $f(1 \mid M=1) \ge 0$, holding **without additional assumptions**. The suppression statement merely describes typical empirical behaviors of discriminative scores.
>
> We provide distribution histograms in Fig 3 of [link](https://anonymous.4open.science/r/PTDI_r/). This confirms member p-values concentrate near 0 with low density near $p=1$, validating the condition in practice.
> When members do appear near $p=1$ (Zlib score in Fig 3(b)), it simply results in **more conservative FIR control**, as JKBB conservatively overestimates the null proportion (Fig 1).
>
> [1] Selection by prediction with conformal p-values. (JMLR 2023)
>
> [2] Testing for outliers with conformal p-values. (AoS, 2021)
>
> [3] Membership inference attacks cannot prove that a model was trained on your data. (SaTML 2025)

---

> > ### Author Rebuttal · Reviewer_3Gd6 · 2026-04-04
> >
> > Thank you for the detailed rebuttal. The clarifications on the empirical setup including confidence intervals, the correction of the figure inconsistency, and the discussion of the theoretical assumptions are helpful and improve my understanding of the paper.
> > While I still have some reservations regarding the strength of the empirical evidence and the positioning with respect to prior work in conformal prediction, I believe these concerns are now better clarified and could be addressed in a revision.
> > Overall, the rebuttal positively updates my assessment, and I will revise my score accordingly.

---

> > > ### Author Response · Authors · 2026-04-04
> > >
> > > Thank you for the thoughtful response and positive feedback. We are glad that our rebuttal helped clarify several important aspects of the paper. We appreciate these helpful comments and will clarify these points more clearly in the final version.
> > >
> > > Thank you again for the constructive engagement throughout this process and for your willingness to **positively update your score**. We truly appreciate your time and valuable guidance.

---

### Decision · Program_Chairs · 2026-04-30

**Decision:**

Accept (regular)

**Comment:**

This work applies conformal prediction techniques to provide provable False Identification Rate control in the task of training data identification for LLMs. It makes a solid contribution by uncovering a new, well-motivated application of conformal prediction. The proposed method is theoretically well-founded and practically applicable. There are concerns that the proposed framework assumes no distribution shift between calibration and test data. Reviewers have provided comments and suggestions about improving the empirical results. I trust the authors to implement the revision promised in the rebuttal. I recommend acceptance.